# Learning from Synthetic Labs: Language Models as Auction Participants

## Abstract

This paper investigates the behavior of simulated AI agents (large language models, or LLMs) in auctions, introducing a novel synthetic data-generating process to help facilitate the study and design of auctions. We find that LLMs reproduce well-known findings from experimental literature in auctions across a variety of classic auction formats. In particular, we find that LLM bidders produce results consistent with risk-averse human bidders; that they perform closer to theoretical predictions in obviously strategy-proof auctions; and, that in a real-world eBay-style setting, LLMs strategically produce end-of-auction "sniping" behavior. On prompting, we find that LLMs are robust to naive changes in prompts (e.g., language, currency) but can improve dramatically towards theoretical predictions with the right mental model (i.e., the language of Nash deviations). We run 1,000+ auctions for less than $400 with GPT-4o models (three orders of magnitude cheaper than modern auction experiments) and develop a framework flexible enough to run auction experiments with any LLM model and a wide range of auction design specifications, facilitating further experimental study by decreasing costs and serving as a proof-of-concept for the use of LLM proxies.

## 1 Introduction

Rapidly advancing capabilities of large language models (LLMs) raise an exciting possibility: might LLM agents serve as cost-effective proxies for human bidders, thereby making experiments orders of magnitude cheaper to run? In this paper, we contribute to an emerging literature on LLM proxies by testing whether LLM agents can replicate key empirical regularities of the auctions literature. We map LLM behavior in standard auction formats against established human and theoretical benchmarks from economics, providing a proof-of-concept for low-cost, large-scale auction experimentation using LLM bidders.

In Section 3.1, we begin by benchmarking play with LLM bidders in classic environments: sealed-bid auctions in independent, private-value settings. When there are departures from the theory, we are interested in whether the departures agree with empirical results (Kagel and Levin, 1993; Kagel and Roth, 2020). First, we find that agents fail to replicate revenue equivalence results (as existing empirical evidence suggests). We find that bids in the SPSB auction are higher than bids under the FPSB auction, as would be expected, but that there is a smaller separation between the two than predicted by theory. This is primarily due to bids under the FPSB auction being higher than the risk-neutral Bayes-Nash equilibrium suggests. One possible explanation is that LLMs play according to some level of risk aversion: this is consistent with Cox et al. (1988)'s evidence from over 1,500 IPV auction experiments, which finds that revenue under the FPSB auction is higher than under the SPSB auction and suggests that this is due to risk aversion.

Next, we fix the second-price auction and consider classic ways to make the auction 'easier' to play, to learn whether LLMs exhibit behavior similar to humans in the face of cognitive constraints. Theoretical and experimental studies (Li, 2017) argue that ascending clock formats are less cognitively demanding than sealed-bid ones, causing humans to make less mistakes relative to the dominant strategy of bidding one's value. In Section 3.2, we find that LLMs also find clock formats easier to play – they are more likely to exit the auction (the analogue to bidding in the sealed-bid format) at their value. Following Li (2017), we study this in an affiliated private value setting, replicating their results with LLMs. We also compare the typical clock auction with a blind variant (a clock auction

where bidders do not know when others drop out) and the corresponding second-price sealed-bid auction, and find that switching to a clock format accounts for most of the improvement in LLM play. For robustness, we also run these experiments in an IPV setting and obtain the same results.

Both of these classic results are some of the reasons economists have found auction theory to be beautiful over the last few decades, but auctions can also be useful (Milgrom and Segal, 2020). Auctions are used to clear billions of dollars annually, and one may ask how LLMs fare in less controlled settings. To answer this, we design a more realistic auction environment inspired by eBay's online marketplace. In Section 4, we examine two design parameters: a hidden reserve price and a modified closing rule (soft-close, which extends the auction when there is bidding activity). First, we demonstrate that LLM agents replicate emergent, real-world bidding behaviors seen on eBay with *no* additional prompting: in auctions with a hard closing time, LLMs frequently delay bids until the last second to 'snipe' their competitors as humans would. Second, experiments testing the design of an extended closing rule greatly reduce bid sniping and improve price discovery, echoing dynamics of online auctions between Amazon and eBay in the early 2000s (Roth and Ockenfels, 2002).

Finally, the ability to interface more directly with our experimental subjects also enables us to study mechanism 'framing' more carefully than with human agents. After all, auctions are most prized as mechanisms to allocate goods efficiently, but if agents don't understand the rules they can't be expected to play well. In Section 5, we conclude the body of our paper by testing six interventions designed to influence LLM bidders' understanding of the underlying economics of the setting and their willingness to follow rules. We find that all the interventions designed to improve the correctness of an LLM agent's bidding strategy did so, with the logic of Nash deviations helping the most. Unsurprisingly, an intervention where we incorrectly reveal a false statement (that the SPSB's dominant strategy is to bid half of one's value) causes LLMs to play poorly, but interestingly, the LLM continues to experiment with new strategies upon determining the given advice is bad.

In the Appendix we report complete simulation procedures, prompts and numerous robustness checks. In particular, we ran experiments with prompts from other contemporaneous auctions experiments (e.g., the Appendix script from Li (2017)), with prompts in different languages and with different currencies, and with varying numbers of bidders. All experiments are run with chain-of-thought agents, and we also run an ablation study with out-of-the-box agents.

To obtain the data for these empirical results, we have developed a code repository to systematically run experiments with a certain number of bidders and a particular prompt. In particular, our repository is flexible enough that it can be used to generate synthetic data for almost any describable auction format and including auctions with single or multiple goods.[1] For the experiments described in this paper, we ran more than $1,000$ auctions with more than $5,000$ GPT-4o agent participants for costs totaling less than \$400.

## 1.1 RELATED WORK

**Auctions:** There's a vast quantity of theoretical and experimental literature on auctions. While Krishna (2009)'s textbook and Kagel and Roth (2020)'s handbook provided invaluable general resources, we will stay focused only on the citations relevant to the results in this paper.

From classic theory, the benchmark of revenue equivalence in the risk-neutral case is exposited in the seminal Myerson (1981). The experimental evidence for departures due to risk-aversion in the FPSB auction are thoroughly documented in Coppinger et al. (1980) and Cox et al. (1988)'s survey. The experimental evidence for the common error of bidding above one's value in the SPSB auction is documented in Kagel and Levin (1993).

Further, recent work on obvious strategy-proofness began with Li (2017), who demonstrates empirically that human subjects tend to be less truthful in second price sealed bid auctions than ascending clock auctions in the APV setting, even though the two auctions are strategically equivalent. Li also provides a theoretical framework for the results. To better understand our simulations, we also consider the experimental evidence presented by Breitmoser and Schweighofer-Kodritsch (2022), who investigate intermediate auction formats that decompose the behavioral effects in Li (2017).

---

[1]On publication, we will make the code-base public in the hopes it facilitates additional research into the role of LLMs as modeling human bidder behavior.

**LLMs as simulated agents:** Recent LLMs, having been trained on an enormous corpus of human-generated data, are able to generate human-like text and reasoning (Achiam et al., 2023; Bubeck et al., 2023). Yet, they are far from perfect – in particular, displaying limited planning abilities and reflecting various cognitive biases endemic to human agents (Wan et al., 2023).

There is a growing literature on using these human-like AI models as simulated agents in economics and social science studies (Aher et al., 2023; Park et al., 2023; Brand et al., 2023). In this literature, Horton (2023) replicates four classical behavioral economics experiments by endowing a single LLM agent with different personas and querying it about its decisions and preferences. Manning et al. (2024) enable multiple GPT-4 agents to interact and simulate various social science scenarios, including bargaining, job interviews, and bail-setting. Finally, Manning and Horton (2025) and Raman et al. (2024) benchmark the ability of LLM agents to conduct play over a broad range of games and tasks.

**LLMs in auctions:** Some existing work has already begun using LLMs as simulated agents in auction experiments. Fish et al. (2024) study collusive behavior in first-price sealed-bid auctions between two LLM agents, with the LLM as price setter. Chen et al. (2023) study how to produce super-human play in auctions. Finally, Manning et al. (2024) runs a limited study an a variant of an open-ascending clock auction with three bidders, focusing on deviations from rational economic theory in considering bidders' values and the final clearing price.

## 2 METHODS

### 2.1 LLM AS AUCTION PARTICIPANTS

We use a single LLM API call to represent one bidder in our auction simulations. Unlike traditional agent-based modeling, we do not anchor bidder behaviors or personas mechanically beyond a random name to index them (e.g., Alice, Betty or Charles). To ensure that LLM agents make profit-maximizing decisions, we append a prompt prefix that explicitly instructs them to seek long-run profit maximization. This instruction is universal in all the experiments ran in this paper, and is inspired by (Fish et al., 2024; Chen et al., 2023). Complete details on prompts and execution can be found in Appendix B.1.

Through the paper, we utilize GPT-4o as the core agent model. This choice is motivated by three reasons: (1) the literature we are in conversation with (Horton (2023); Shao et al. (2023); Wang et al. (2025)) primarily uses this model, (2) the model is cheap and ubiquitous, and (3) the model transparently does not have reasoning built in (enabling us to control the chain-of-thought specification, as reported in Appendix B.1). For robustness, we also report the results from GPT-o3, a well-understood 'reasoning' model in Appendix C.

### 2.2 SIMULATION PROCEDURE

For each auction experiment, we simulate $n$ (often 3) LLM agents bidding against one another. Each setting is repeated multiple times, with randomized private values drawn for each round. We design our multi-round protocol to mirror standard laboratory experiments in the auction literature. Full prompt texts are in Appendix I, and an example round is detailed in Appendix B.1. API calls are made using the Python package Expected Parrot Domain-Specific Language (EDSL), a wrapper that structures LLM queries and responses in the desired format (Horton et al., 2024). For every auction, we use GPT-4o at a temperature of $0.5$.[2]

## 3 BENCHMARKING AGAINST EXISTING AUCTIONS

### 3.1 SEALED BID AUCTIONS WITH INDEPENDENT PRIVATE VALUES

We begin by examining the First-Price Sealed-Bid (FPSB), Second-Price Sealed-Bid (SPSB) auctions in an independent private values (IPV) setting.

---

[2]We set a non-zero temperature to encourage diversity in strategies and reflections. No parameter tuning was performed to achieve the results of this paper.

### 3.1.1 SETTING

There are three bidders in each auction, and each bidder $i$'s value is drawn from an independent, uniform distribution $v_i \sim U[0, 99]$. Bidders, upon observing their value, submit a sealed bid $\beta(v)$, with $\beta(\cdot)$ a vector mapping each component value to its corresponding bid, corresponding to a strategy profile for reporting based on the valuation profile. In the FPSB auction, the highest bidder pays her bid and receives the prize (and all other bidders pay 0 and receive no prize). In the SPSB auction, the highest bidder pays the second-highest bid and receives the prize (and all other bidders pay 0 and receive no prize). Bids are submitted in \$1 increments and ties are resolved randomly.

### 3.1.2 THEORETICAL AND EMPIRICAL BENCHMARKS

The risk-neutral equilibrium strategy in both of these auction formats is well understood.

In the FPSB auction, the risk-neutral Bayes-Nash equilibrium takes the form: $\beta_i(v_i) = \frac{n-1}{n}v_i$. In the case of FPSB auctions, experimental evidence consistently finds bids above the risk-neutral BNE prediction – this is commonly identified as evidence of bidder risk-aversion. In an FPSB auction, bidding higher increases the probability of winning but decreases the potential profit if one does win. Risk-averse bidders may be willing to accept lower potential profits in exchange for a higher chance of winning, leading to bids above the risk-neutral BNE.

In the SPSB auction, the equilibrium holds in dominant strategies: $\beta_i(v_i) = v_i$. Similarly, experimental data for SPSB auctions also reveals evidence of overbidding. Kagel et al. (1987) and Kagel and Levin (1993) famously provide of evidence this behavior, in particular highlighting that a significant proportion of participants (typically around 60-70%) in SPSB auctions submitted bids above their true values. However, given the SPSB has an equilibrium in dominant strategies the literature classifies this 'mis-play' as evidence of alternate behavioral mistakes. These findings highlight the complexity of human behavior in auction settings. Several explanations have been proposed, including misconceptions about the second-price rule (Kagel et al., 1987) and strategic uncertainty or beliefs about other bidders' irrationality (Crawford and Iriberri, 2007).

### 3.1.3 SIMULATION EVIDENCE

For each of the IPV auction formats, we perform 5 experiments with 15 rounds and 3 LLM bidders competing over a prize.[3] First, we see that the sealed-bid formats demonstrate evidence of monotone bidding, which is one of the most stable hallmark of experimental lab results on auctions.

For the FPSB, the Loess-smoothed data curve matches the BNE prediction. However, this curve has high variance, with a substantial mass of bidders bidding between their value and the BNE prediction – this mass corresponds to the classic evidence of risk-aversion (Cox et al., 1988).

As with human bidders, LLM bidding in the SPSB auction exhibits numerous departures from equilibrium play. In our experiments, 37.78% of bids match bidders' values, closely aligning with the 27% reported by Kagel and Levin (1993).[4]

However, the nature of these deviations differs strikingly. In a SPSB auction with 5 players, Kagel and Levin (1993) find the distribution of bids to be right-skewed, with most of the bidding mass (67%) on over-bidding one's value. In contrast, we find the distribution of bids to be primarily left-skewed, with most of the bidding mass (60%) on under-bidding one's value. The contrast highlights that despite both under-bidding and over-bidding being violations of strategy-proofness, they represent two distinct types of behavioral errors empirically – an insight that has not been well rationalized by the auctions literature yet. The prevalence of underbidding among LLM bidders further suggests loss-aversion to a degree not observed in human bidders, and suggests an interesting avenue for future research characterizing LLMs as economic actors: namely, that LLMs may naturally ex-

---

[3]While these specifications are designed to meet the 'classic' auction formats of the literature, we also report the results of several variants in Appendix E. In particular, we run one-shot auction simulations (as opposed to multi-round) with $n = 4, 5$, and 10 bidders, auctions with prompts using various currencies, and auctions with non-English prompts using various currencies.

[4]Kagel and Levin (1993) define value-bidding as bidding within \$0.05 of one's value (with \$0.01 bid increments), while we define value-bidding as bidding within \$1 of one's value (with \$1 bid increments).

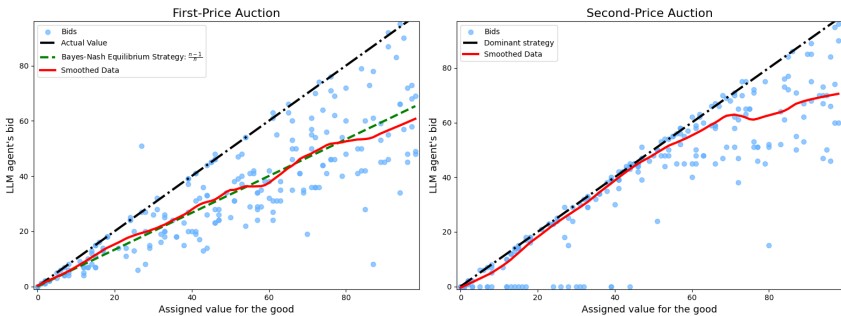

Figure 1: **Comparison of FPSB(Left) and SPSB(Right) under IPV setting.** LLM bids are represented by blue circles. The black dotted line represents the 45° line (the dominant strategy for SPSB auctions), the green dashed line represents the Bayes-Nash Equilibrium on the FPSB plot, and the solid red line represents the LOESS-smoothed bidding curve for both plots. We plot LOESS curves to impose no parametric assumptions (e.g., monotonicity) on bidding behavior.

hibit *more* loss-aversion than human actors. Taken as a whole, we see these mistakes in the SPSB as additional motivation for exploring behavioral phenomena in LLMs under economic mechanisms.

### 3.2 OBVIOUSLY STRATEGY-PROOFNESS

For the rest of the paper, we restrict our attention to the second-price auction mechanism. Having considered LLM play in a variety of sealed-bid auctions, we now vary the representation of the second-price auction to make the auction 'easier' to play. Namely, we consider clock auctions.

#### 3.2.1 SETTING

There are three bidders in each auction but now, following Li (2017), bidders draw *affiliated private values* (APV) of the form $v_i = c + p_i$. The common component $c$ is drawn once, $c \sim U[0, 20]$, and the private component $p_i$ is drawn $n$ times independently, $p_i \sim U[0, 20]$. Winners of the auction receive their own value of the prize $v_i$ when they win, and the 'common' and 'private' components serve to make values correlated. We switch to considering APV settings in this section for two reasons. First, in spirit of our larger goal to provide many benchmarks of LLM play in various environments. Second, to enable comparison with the existing empirical literature; the two largest experiments of which have been conducted in APV settings.

The *ascending clock auction* (AC) has a global price clock that increases in some fixed increment and bidders who choose when to drop out. The last bidder remaining wins the prize at the price where the second to last bidder dropped out. If multiple bidders are the last to drop out, we resolve ties by randomly allocating the prize among them and setting the winning price to that mutual drop out price. We also study the *blind ascending clock auction* (AC-B), a variant where bidders are not told when other bidders drop out.

#### 3.2.2 THEORETICAL BENCHMARKS

What effect an affiliated values environment has on strategic behavior may not be ex-ante obvious. On one hand, the correlation in values may strengthen incentives for extremal-valued bidders, so that extremal-valued bidders are actually *more* likely to bid their value even in the sealed-bid format (echoing arguments from Li (2017)). On the other hand, APV may be cognitively more complex for bidders relative to IPV environments, thereby inducing errors in play. In either case, the three auction formats in this section are strategically equivalent, and so the affiliation is, in the rational sense, a red herring: for all three auctions it is still a dominant strategy to bid (or drop out at) one's value. The SPSB, AC-B and AC auctions share an equilibrium in dominant strategies:

$$\beta^{**}(v) = v.$$

An agent's bid in the AC format is equivalent to the price at which they drop out of the auction at. We will use the terms 'bid' and 'drop-out price' interchangeably in the context of clock auctions.

### 3.2.3 EMPIRICAL BENCHMARKS AND RESULTS

For each of the APV auction formats, we perform 5 experiments with 15 rounds and 3 LLM bidders competing over a prize. A single round for the clock format is the run of a global clock from the initial price of \$0 to the final clearing price. Complete detail for the simulation procedure of a single clock round is provided in B.2.

Our main results are plotted in Figure 2. We find that the clock formats induce play significantly closer to one's value than the sealed-bid format. Table 1 summarizes this concisely: the clock formats, aggregated, have an average difference between bid and value of $1.83$ (with $73.9\%$ of agents dropping out at their value, excluding winners), while the SPSB bidding has an average difference between bid and value of $6.26$ (with $18.7\%$ of all bids equal to an agent's value) [5]. This corresponds closely with Li (2017)'s experimental results: aggregated across all rounds, he reports a mean difference between bid and value of $2.48$ for the ascending clock auction, and a mean difference of $6.52$ for the sealed-bid auction. In general, the clock format improves play for LLM bidders in much the same way as with human bidders.

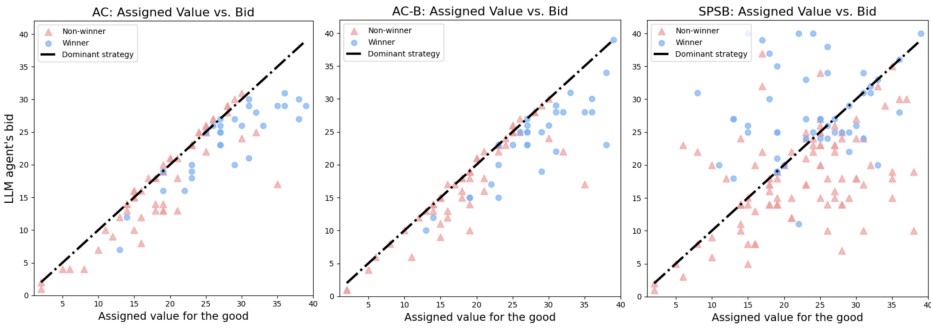

Figure 2: **Comparison of three strategically equivalent auctions (the AC, AC-B and SPSB auctions) in an APV setting.** The AC and AC-B auctions are obviously strategy-proof, whereas the SPSB auction is only strategy-proof. The black dot-dash line represents the dominant strategy of value-bidding. In the clock auctions, red triangles indicate non-winners' drop out prices and blue circles indicate the final price that winners win at. In the SPSB auction, red triangles indicate non-winners' bids and the winning bid in clock auctions.

| Metric | AC | AC-B | SPSB |
|---|---|---|---|
| Truthful Bidding (%) | 70.0% | 76.7% | 18.7% |
| Average Difference | 2.03 (0.39) | 1.63 (0.40) | 6.26 (0.62) |
| Sample size | 90 | 90 | 150 |

Table 1: Share of truthful bidding in the AC, AC-B and SPSB auctions in an APV setting.

Breitmoser and Schweighofer-Kodritsch (2022) provide additional evidence that clock auctions improve truthful play, and also provide evidence that play under AC is slightly better than under AC-B. While we replicate the former result, we fail to replicate the latter result. Play under the AC auction was not significantly different compared to the AC-B auction ($t = 0.71, p = 0.48$) in our experiments. We report all the two-sample t-tests of the mean absolute difference between bids and values in Table 3 – both AC and AC-B mechanisms show significantly smaller bid-value deviations than the SPSB auction ($p < 0.001$), but not from one another.

Finally, we observe that the clock format improves on a particular failure mode of the SPSB - namely, bidding *above* one's value. The result is that the lion's share of total mistakes in clock auctions come

---

[5]Following the definition in Li (2017), we define truthful bidding in the ascending clock (AC) and ascending clock with bid (AC-B) auctions as dropping out within one increment of the bidder's true value (excluding winners), and in the second-price sealed-bid (SPSB) auction as submitting a bid exactly equal to one's true value.

from dropping out too early rather than too late. Although OSP mechanisms cannot rationalize a difference between over- and under-bidding (both represent deviations from an obviously dominant strategy), this evidence further suggests that over- and under-bidding represent categorically *distinct* failure modes. Further exploration into rationalizations of the difference between these two failure modes remains an interesting and open question for future research.

## 4 EBAY AUCTION DESIGN

Having tested LLMs against classic results in auction theory, we now build an auction environment to test components of auction design in a setting inspired by eBay's online marketplace. eBay's auction platform, studied by Einav et al. (2011) and others, features various design elements – for example, a hidden reserve price (as opposed to a reserve price) and proxy bidding – that are *in theory* less critical to auction outcomes yet *in practice* appear to have substantial effects on bidder behavior. These differences highlight the gap between textbook theory and real-world bidding, where seemingly small design details can alter bidder behavior and thus price discovery and revenue.

One dramatic departure from simple, truthful bidding observed in practice is 'bid sniping,' or the practice of placing bids in the final moments of an auction to avoid triggering competitive responses. Sniping is prevalent on eBay, with dedicated e-snipe services automating last-second bid submissions.[6] One proposed avenue to curb sniping is the 'closing rule' (also called a 'soft-close' or 'auction extension' rule) – namely, when a new maximum bid comes in at the last minute, extend the auction close to allow for more bidding. This design detail has been credited with the substantial difference in bidding dynamics between earlier auctions on Amazon.com, which had this auction-extension rule, and auctions on eBay, which do not. Empirically, Roth and Ockenfels (2002) and Ockenfels and Roth (2006) show that much less sniping occurred on Amazon.com (with a closing rule) compared to eBay (with no closing rule).

In this section, we simulate a stylized version of the eBay auction design to consider two design parameters more closely: the hidden reserve price and the implementation of such a closing rule.

### 4.1 SETTING

In each eBay simulation, there are three bidders. Without the modified closing rule, the auction runs for 10 bidding periods. Bidders draw private values uniformly and independently from $[0, 99]$. The item to be bid in standard auctions is specified as a "prize". In eBay auctions, we arbitrarily set the prize item as a "256GB IPhone 16 pro" and the item condition is set to be "used" for realism. In regard to the auction environment, we consider four treatments, varying the hidden reserve and the closing rule: **(T1:)** a standard eBay auction with proxy bidding; **(T2:)** an eBay auction with a modified (auction extension) closing rule; **(T3:)** a standard eBay auction with a hidden reserve price; and **(T4:)** an eBay auction with both a modified (auction extension) closing rule and a hidden reserve price.

For each hidden reserve price design, we run experiments with hidden reserves at $r = 40, 50$, and $60$. With this stylized eBay framework we abstract away from additional eBay design features such as shipping fees and Buy-It-Now.

### 4.2 SIMULATION PROCESS OF EBAY AUCTION

For eBay auctions, we discretize the continuous bidding period into 10 periods. Each period, LLM agents decide whether to increase their bid or hold their current bid. The bidding process follows a structured format, where bidders act in a predefined sequence each period (e.g., Charles, then Alice, then Betty). In particular, agents are informed of past price changes through a transcript, such as "On day 1, the price changed to 1. On day 2, the price changed to 3." The complete simulation process is reported in Appendix B.3 and prompts are listed in Appendix I.

---

[6]E.g., https://www.esnipe.com/.

## 4.3 SIMULATION EVIDENCE

**Hidden reserve price results.** Figure 10 shows that the auction revenue remains basically unchanged regardless of the experiment type. In Appendix Table 4, we report a table of two-sided t-tests making this explicit.

**Closing rule results.** Second, we demonstrate that modifying eBay's closing rule to allow for an extension of the auction when there is final-period bidding has a major effect on bidder behavior. We define the final-winning-bid time as the last period in which the eventual winner changes her maximum bid. If she raises her maximum in a later period, that later period becomes the recorded time. This is a measure for when the bidder who would win the auction was determined.

Figure 3 reports a CDF across the four treatment types (T1 - T4). Without the modified closing rule, we see that LLMs predominantly bid in the period before close; however, with the modified closing rule, this behavior disappears. In effect, the modified closing rule shifts the winning bid time earlier in clock time, so that the auction's winner is chosen earlier (thereby improving auction efficiency). For further visualization of this dynamic, Figure 9 in the Appendix reports the 'final winning bid time' as a histogram across auction instances with and without the modified closing rule.

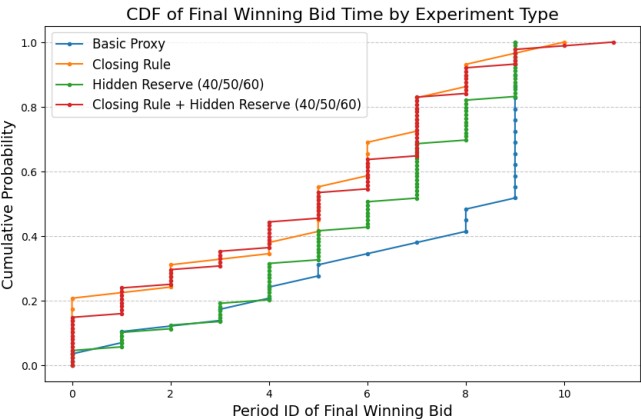

Figure 3: **CDF of Final Winning Bid Timing Across Variations on eBay-Style auctions, IPV environment.** The cumulative distribution function of the final winning bid time for each auction format. Formats with a modified closing rule brings earlier, both with and without the hidden reserve

## 5 INTERVENTION DESIGN IN SEALED-BID AUCTION

It matters how agents perceive mechanisms. Here, we present six interventions which affect the *perception* of the SPSB auction without affecting underlying theoretical optimality (given the SPSB's equilibrium in dominant strategies). Of the six interventions, the first three seek to prime agents on particular conceptual frameworks, the next two provide (good and bad) advice, and the last is a behavioral intervention.

The detailed intervention designs are: **Menu-Description (menu_intervention)**, which provides the extensive form of the auction as a menu to the agent (inspired by Gonczarowski et al. (2023)); **Clock-Description (proxy_intervention)**, which describes the ascending-clock process—though the proxy clock is introduced before the game begins, the auction itself is still operated as an SPSB auction (inspired by Breitmoser and Schweighofer-Kodritsch (2022)); **Nash-Deviation (nash_intervention)**, which illustrates a Nash-style thought process by asking what happens if one bids up or down; **Direct Revelation (dominant_strat_intervention)**, which directly reveals the dominant strategy of bidding one's value; **Wrong Direct Revelation (wrong_strat_intervention)**, an ablation study that incorrectly instructs the agent to bid half of its value; and **Risk-neutrality prompting (risk_intervention)**, which defines risk-neutrality in terms of gambles and explicitly prompts the LLM to act in a more risk-neutral manner. The full intervention prompts are provided in Appendix J and are appended to the end of the rule explanation prompt when eliciting LLM bids.

## 5.1 SIMULATION EVIDENCE

The results of the six interventions are reported below. To characterize the error from the theoretical prediction at `bid = value`, we also report the $R^2$ against $\beta^{**}(v) = v$ in Table 2. However, as was clear in the discussion of the empirical literature in Section 3.1, overbidding and underbidding constitute different failure modes – in particular, many human bidders suffer the failure mode of overbidding in the SPSB auction.

| | Intervention Type | $R^2$ | SS Above | SS Below | Prop Above | Prop Below |
|---|---|---|---|---|---|---|
| | No Intervention | 0.4845 | 7,213 | 78,652 | 0.0840 | 0.9160 |
| **Priming** | Menu | 0.5337 | 11,104 | 57,685 | 0.1614 | 0.8386 |
| | Proxy | 0.2966 | 177 | 85,488 | 0.0021 | 0.9979 |
| | Nash | 0.8365 | 1,160 | 24,591 | 0.0450 | 0.9550 |
| **Advice** | Dominant Strategy | 0.9533 | 2,665 | 5,521 | 0.3256 | 0.6744 |
| | Wrong Strategy | -0.8258 | 169 | 141,137 | 0.0012 | 0.9988 |
| **Behavioral** | Risk | 0.4004 | 59 | 89,920 | 0.0007 | 0.9993 |

Table 2: **Intervention Type Analysis by Category, SPSB and IPV environment.** The table compares how each intervention changes bidding performance relative to the dominant-strategy benchmark $\beta^{**}(v) = v$. Larger $R^2$ means a tighter overall fit, while 'SS Above/Below' and 'Prop Above/Below' decompose the error into classic over- versus under-bidding. Priming agents to consider the logic of Nash deviations via prompt cuts both types of error (raising $R^2$ from 0.48 to 0.84), but the direct-advice treatment is even stronger, pushing $R^2$ above 0.95 and halving the under-bidding sum-of-squares. In contrast, misleading advice ('Wrong Strategy') produces the worst outcomes, yielding a negative $R^2$ and almost universal under-bidding. Finally, the risk-neutrality prime largely eliminates over-bidding yet leaves systematic caution below the 45° line.

As such, we also decompose the $R^2$ into the sum of squares above and below $\beta^{**}(v) = v$ to report each intervention's performance in terms of the two components of overbidding and underbidding. We report each of these interventions alongside the results from the original SPSB experiments of Section 3.1 (No Intervention) in Table 2.

Unsurprisingly, the 'advice' intervention which reports the true dominant strategy improves play the most, with an $R^2 = .95$. However, interestingly, the second most effectively intervention is the Nash intervention. It makes significant improvements relative to base play in the SPSB by both reducing the magnitude of over-bidding and under-bidding – in both cases inducing agents to play *closer* to their true value. For plots of these interventions, we report Figure 11 in the Appendix.

## 6 CONCLUSION

This paper reports the results of more than 1,000 auction experiments with GPT-4o participants. In particular, we find behavior that conforms with important experimental results with humans (from both in the lab and in the field). We see the main contribution of this work as establishing a framework for considering LLM evidence as proxy for human evidence in mechanism design.

In particular, while this paper focuses on auction design, future work may use LLM sandboxes to test other kinds of economic mechanisms; e.g., voting, matching, contract design, etc. As LLM models are increasingly validated as proxies for human behavior, LLM agents can then be used to obtain what would otherwise be prohibitively expensive evidence. As a provocative example, while ethical and financial constraints make it impossible to run voting experiments at the scale of nations, it may be possible to run such experiments with LLM agents.

Finally, we are particularly interested in the use of LLM-based techniques to generate synthetic data that is useful for informing economic design. In particular, some auction formats, such as combinatorial auctions, are complex and difficult to run frequently and at scale in traditional lab experiments. Augmenting these traditional lab experiments with LLM-agent experiments, when correctly validated, may open up wide new avenues to better understand the design tradeoffs in these kinds of complex and often high-stakes environments.

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

# APPENDIX

## A PROMPTS

### A.1 INDEPENDENT PRIVATE VALUES PROMPTS

Following the simulation procedure outlined in Section 2.2, we employ the following prompts for the auctions tested. For each prompt, only the third paragraph (describing the allocation rule and payment) differ. For the FPSB, we describe the auction format as follows:

```
    In this game, you will participate in an auction for a
        prize against {{num_bidders}} other bidders. You
        will play this game for {{n}} rounds.
    At the start of each round, bidders will see their
        value for the prize, randomly drawn between \$0
        and \${{private}}, with all values equally likely.
```

```
After learning your value, you will submit a bid
    privately at the same time as the other bidders.
    Bids must be between \$0 and \${{private}} in \${{
    increment}} increments.
The highest bidder wins the prize and pays their bid
    amount. If you win, your earnings will increase by
     your value for the prize, and decrease by your
    bid. If you don't win, your earnings will remain
    unchanged.
After each auction, we will display all bids and the
    winner's profits. Ties for the highest bid will be
     resolved randomly.
```

For the SPSB, we describe the auction format as follows:

```
In this game, you will participate in an auction for a
     prize against {{num_bidders}} other bidders. You
    will play this game for {{n}} rounds.
At the start of each round, bidders will see their
    value for the prize, randomly drawn between $0 and
     ${{private}}, with all values equally likely.
After learning your value, you will submit a bid
    privately at the same time as the other bidders.
    Bids must be between $0 and ${{private}} in ${{
    increment}} increments.
The highest bidder wins the prize and pays the second-
    highest bid. If you win, your earnings will
    increase by your value for the prize, and decrease
     by the second-highest bid. If you don't win, your
     earnings will remain unchanged.
After each auction, we will display all bids and the
    winner's profits. Ties for the highest bid will be
     resolved randomly.
```

For the TPSB, we describe the auction format as follows:

```
In this game, you will participate in an auction for a
     prize against {{num_bidders}} other bidders. You
    will play this game for {{n}} rounds.
At the start of each round, bidders will see their
    value for the prize, randomly drawn between $0 and
     ${{private}}, with all values equally likely.
After learning your value, you will submit a bid
    privately at the same time as the other bidders.
    Bids must be between $0 and ${{private}} in ${{
    increment}} increments.
The highest bidder wins the prize and pays the third-
    highest bid. If you win, your earnings will
    increase by your value for the prize, and decrease
     by the third-highest bid. If you don't win, your
    earnings will remain unchanged. If there are less
    than three bids, no one will win the auction.
After each auction, we will display all bids and the
    winner's profits. Ties for the highest bid will be
     resolved randomly.
```

For the all-pay auction, we describe the auction format as follows:

```
In this game, you will participate in an auction for a
    prize against {{num_bidders}} other bidders. You
    will play this game for {{n}} rounds.
At the start of each round, bidders will see their
    value for the prize, randomly drawn between $0 and
    ${{private}}, with all values equally likely.
After learning your value, you will submit a bid
    privately at the same time as the other bidders.
    Bids must be between $0 and ${{private}} in ${{
    increment}} increments.
The highest bidder wins the prize. All bidders (
    including the winner) pay their submitted bid. If
    you win, your earnings will increase by your value
     for the prize, and decrease by your bid. If you
    don't win, your earnings will still decrease by
    your bid.
After each auction, we will display all bids and all
    bidders' profits. Ties for the highest bid will be
     resolved randomly.
```

## A.2  AFFILIATED VALUES PROMPTS

The affilate-value prompts follows the style of the last section's prompts closely. The only difference is in the first two paragraphs, where agents are informed of the APV environment. For the SPSB, we describe the auction format as follows:

```
In this game, you will participate in an auction for a
    prize against {{num_bidders}} other bidders. You
    will play this game for {{n}} rounds.
At the start of each round, bidders will see their
    value for the prize. Your value for the prize will
     be calculated as follows:
1. First we will randomly draw a common value between
    {{common_low}} and {{common_high}}, with all
    values equally likely.
2. Then, for each bidder, a private taste adjustment
    will be drawn between 0 and {{private}}, with all
    values equally likely.
Your value for the prize is equal to the common value
    plus your private taste adjustment. You will not
    learn the common value or your private taste
    adjustment separately. This means that each person
     in your group may have a different value for the
    prize. However, if you have a high value, it is
    more likely that other people in your group have a
     high value.
After learning your value, you will submit a bid
    privately at the same time as the other bidders.
    Bids must be between $0 and ${{common_high +
    private}} in ${{increment}} increments.
The highest bidder wins the prize and pays the second-
    highest bid. If you win, your earnings will
    increase by your value for the prize, and decrease
     by the second-highest bid. If you don't win, your
     earnings will remain unchanged.
After each auction, we will display all bids and the
    winner's profits. Ties for the highest bid will be
     resolved randomly.
```

The implementation of the clock auction simulates a global clock price, with prices beginning at $0$ and increasing in $1 increments. The clock auction format descriptions closely mirror the SPSB format above, but with the added description of the clock mechanism in the fourth and fifth paragraphs. In the AC auction, we describe the auction format as follows:

```
In this game, you will participate in an auction for a
    prize against {{num_bidders}} other bidders. You
    will play this game for {{n}} rounds.
At the start of each round, bidders will see their
    value for the prize. Your value for the prize will
     be calculated as follows:
1. First we will randomly draw a common value between
    {{common_low}} and {{common_high}}, with all
    values equally likely.
2. Then, for each bidder, a private taste adjustment
    will be drawn between 0 and {{private}}, with all
    values equally likely.
Your value for the prize is equal to the common value
    plus your private taste adjustment. You will not
    learn the common value or your private taste
    adjustment separately. This means that each person
     in your group may have a different value for the
    prize. However, if you have a high value, it is
    more likely that other people in your group have a
     high value.
The auction proceeds as follows: First, you will learn
     your value for the prize. Then, the auction will
    start. We will display a price to everyone in your
     group that starts at 0 and counts upwards in {{
    increment}} USD increments, up to a maximum of {{
    common_high + private}}. At any point, you can
    choose to leave the auction, and anytime a bidder
    leaves, we will broadcast that information to all
    the remaining bidders.
When there is only one bidder left in the auction,
    that bidder will win the prize at the current
    price. If you win, your earnings will increase by
    your value for the prize, and decrease by the
    current price. If you don't win, your earnings
    will remain unchanged.
After each auction, we will display all bids and the
    winner's profits. Ties for the highest bid will be
     resolved randomly.
```

The AC-B auction description closely resembles the AC format, but with the explicit caveat that we will not notify bidders when competitors drop out. We describe the auction format as follows:

```
In this game, you will participate in an auction for a
    prize against {{num_bidders}} other bidders. You
    will play this game for {{n}} rounds.
At the start of each round, bidders will see their
    value for the prize. Your value for the prize will
     be calculated as follows:
1. First we will randomly draw a common value between
    {{common_low}} and {{common_high}}, with all
    values equally likely.
2. Then, for each bidder, a private taste adjustment
    will be drawn between 0 and {{private}}, with all
    values equally likely.
```

```
Your value for the prize is equal to the common value
    plus your private taste adjustment. You will not
    learn the common value or your private taste
    adjustment separately. This means that each person
     in your group may have a different value for the
    prize. However, if you have a high value, it is
    more likely that other people in your group have a
     high value.
The auction proceeds as follows: First, you will learn
     your value for the prize. Then, the auction will
    start. We will display a price to everyone in your
     group that starts at 0 and counts upwards in {{
    increment}} USD increments, up to a maximum of {{
    common_high + private}}. At any point, you can
    choose to leave the auction, but we will not tell
    any bidder when someone leaves.
When there is only one bidder left in the auction,
    that bidder will win the prize at the current
    price. If you win, your earnings will increase by
    your value for the prize, and decrease by the
    current price. If you don't win, your earnings
    will remain unchanged.
After each auction, we will display all bids and the
    winner's profits. Ties for the highest bid will be
     resolved randomly.
```

## B    IMPLEMENTATION DETAILS

### B.1    SIMULATION PROCESS IN SEALED-BID AUCTIONS

First, as an overview, a round takes the following form:

1. *Planning.* LLM agents receive a description of the auction format and are asked to propose a bidding plan.

2. *Value Realization.* Each LLM agent is assigned a random private value.

3. *Bidding.* Given their value, each agent places a bid according to their stated plan.

4. *Outcome.* The system computes allocation and payments. Each LLM agent learns its own profit or loss outcome, as well as the full set of bids submitted (but not other agents' private values).

5. *Reflection.* Before the next round, each LLM agent reviews the results and possibly revises its approach. These updates are stored in the HISTORY variable.

6. *Repetition.* One complete "plan-bid-reflect" loop is one round. We repeat 15 rounds in each experiment, with the entire HISTORY carried forward.

During the auction, LLM agents will be first informed about the scenario. After explaining to them the rules and incentives of the aucion, they will be asked to write down a plan for the bidding.

```
You are Bidder Andy.
You are bidding with Bidder Betty, Bidder Charles. {goal}.

{RULE EXPLANATION}

Write your plans for what bidding strategies to test next.
Be detailed and precise but keep things succinct
and don't repeat yourself.
Your plan should be within 100 words.
```

Here, the RULE EXPLANATIONS are specific for each type of auctions. A universal goal prompt -*"Your TOP PRIORITY is to place bids which maximize the your profit in the long run. "* - is included to incentivize the LLM agents and motivate strategic behaviors; they are universal in all the simulations we ran.

The following is the output from Bidder Andy for the first round in a FPSB auction:

> *"I plan to initially explore a strategy where I bid half of my value, then varying my bids between 30-70% of my value in subsequent rounds. Sequentially, I will introduce risk by bidding within +/- 5% of the highest bid from previous round, gathering data about other bidder's tendencies. Lastly, based on gathered data, I'll deploy a more aggressive high bid if losses amass, or low-risk bid if leading, to secure profit. Learning from each round is vital to adapt and refine strategy for profit maximization."*

Once they make the bidding plan, the system will inform the LLM agents their value towards the item in this round and ask them to make a bid according to their plan.

```
You are Bidder Andy.
You are bidding with Bidder Betty, Bidder Charles. {goal}.

{RULE EXPLANATION}

Your value is 73. Your plan is {PLAN}
FOLLOW YOUR PLAN.
How much would you like to bid?
Give your response with a single number and no other texts,
e.g. 1, 44
```

In this example, Bidder Andy valued the item at $73. And according to his plan, he only wanted to bid 30-70% of his value. So, he bid *"36.5"* in this round.

After receiving the bids from all the LLM bidder, the system determines the winner and the profits for each bidders. All the bids and the winner's profit are returned back to the bidders along with whether they win the auction in this round. The part of information is included in the HISTORY variable. For sealed-bid auctions, the HISTORY includes the all the bids, the clearing price, winner's profit, this agent's value, agent's bid and whether they won the auction in that round. An example of the history briefing is provided below:

> *"The previous round histories along with your plans are: In round 0, Your value was 86, you bid 86.0, and your profit was 0.0. Your total profit is 0.0. All the bids for this round were 86.0, 56.0, 21.0. The highest bidder won with a bid of 86.0 and would've preferred to bid 57.0. The winner's profit was 0.0. Did you win the auction: Yes. "*

Before entering the next round, the bidders are asked to do a reflection upon their bidding strategy and the result.

```
You are Bidder Andy.
You are bidding with Bidder Betty, Bidder Charles. {goal}.

{RULE EXPLANATION}

The previous round history is: {HISTORY}

Do a counterfactual analysis of the last round.
REMEMBER that your goal is to win the bid and make higher profits.
LIMIT your OUTPUT within 100 words.
```

An example of Bidder Andy's reflection is provided below. Since he didn't win the auction and incurred a zero profit, his reflection was that bidding up closer to his true value would help him win the game and secure a net profit.

> *"If I bid down by a few dollars, say around $30, I likely still wouldn't have won given the winning bid was $57. I would've merely saved my bid money. If I bid up by $21, matching the winning bid of $57, I may or may not have won due to the random resolution of ties. However, if I'd won, my profit would have been $16, realizing the difference between my value of $73 and my raised bid. So, bidding aggressively could potentially increase my profits in such scenarios. "*

In the following new round, all the bidders are asked to update their plans according to the history of the last round and their reflections.

```
You are Bidder Andy.
You are bidding with Bidder Betty, Bidder Charles. {goal}.

{RULE EXPLANATION}

The previous round history is: {HISTORY}
After careful reflection on previous bidding,
your analysis for last round is {REFLECTION}.

Learn from your previous rounds,
Let's think step by step to make sure we make a good choice.
Write your plans for what bidding strategies to test next.
Be detailed and precise but keep things succinct and don't repeat yourself
LIMIT your plan to 50 words.
```

Here, we show the updated plan for the second round of the same Bidder Andy. After learning from the lessons of bidding only 50% of the value and losing the game, this time the agent updates their plan to the following:

> *"I plan to test higher bidding strategies, around two-thirds of the prize value to foster winning chances. Additionally, if my prize value is extremely high, I will bid aggressively for data-gathering."*

This plan-bid-reflection loop will iterate until the end of the auction. After the last bid, no reflection will be queried. In the IPV setting, there are overall 15 rounds for all the sealed-bid auctions. In the affiliated and common value auctions, there are overall 10 rounds.

## B.2 SIMULATION PROCESS IN ASCENDING CLOCK AUCTIONS

In Ascending Clock Auctions, each round consists of multiple clock cycles, during which every bidder is asked whether they want to stay or drop out at the current clock price. In the first round, bidders are reminded of the auction rules, similar to those outlined in Section B.1. The auction starts with an initial price of 0, which increases incrementally until only one bidder remains or two bidders drop out simultaneously, in which case the winner is chosen randomly. The detailed prompt is listed as follows, with variables enclosed in brackets:

```
You are Bidder Andy.
You are bidding with Bidder Betty, Bidder Charles. {goal}

{RULE EXPLANATION}

Your value towards to the prize is 26 in this round.
The current price in this clock cycle is {current_price}.
The price for next clock cycle is {current_price + increment}.
```

```
        The previous bidding history is: {transcript}.

        {Asking Prompt}
```

To implement plan-bid-reflection within each clock cycle, we modified the Asking Prompt as follows, inspired by Fish et al. (2024):

```
        Do you want to stay in the bidding?
        If you choose yes, you can keep bidding for next clock.
        If you choose No, you will exit and
        have no chance to re-enter the bidding.
        Your response must use these EXACT
        tags below. You must output the ACTION.
        ```
        <PLAN>
        [Write your plans for what bidding strategies to test next.
        Be detailed and precise but keep things succinct
        and don't repeat yourself.
        LIMIT your plan to 50 words. ] </PLAN>
        <ACTION> Yes or No </ACTION>
        <REFLECTION> think about how to justify your choice </REFLECTION>
        ```
```

In AC, if LLM bidders didn't decide to drop out in the first round, they are shown the previous bidding history for the next clock cycle. The history includes the price and how many people had dropped out. And the transcript variable will be the following format:

```
        The previous biddings are:
        ['In clock round 1, the price was 1,
        no players dropped out'].
```

In AC-B, we don't show the previous biddings and in each clock cycle, we directly query LLM agents' decision. So the transcript variable will be None.

A clock auction round ends when only one bidder remains, who is declared the winner, or when two bidders drop out simultaneously, in which case the winner is chosen randomly. Each auction consists of 10 independent rounds with affiliated value settings.

### B.3 SIMULATION PROCESS IN EBAY AUCTIONS

We model continuous time in the synthetic, modified eBay auction environment as follows: each period, bidders bid (i.e., decide whether to increase their maximum bid or hold) in a common knowledge, randomly-permuted order (e.g., today Charles, then Alice, then Betty will bid; tomorrow, Alice, then Betty, then Charles). Moreover, each bidder can decide whether to increase their maximum bid with knowledge of the updated state of the auction based on the actions of bidders that precede them that period. In the final period, bidding is simultaneous and no bidder will know whether they are the last bidder (simulating that, in a continuous time auction, you cannot know if another bidder will get a bid in between you and the auction's close). Under the modified closing rule, if a higher bid is placed in the final period such that there is a new winner, the auction is extended by an additional period. This process continues until no further bids are made.

In particular, for our eBay auctions, we discretize the continuous bidding period into 10 periods. Each period, LLM agents decide whether to increase their bid or hold their current bid. The bidding process follows a structured format, where bidders act in a predefined sequence each period (e.g., Charles, then Alice, then Betty). Agents are informed of past price changes through a transcript, such as "On day 1, the price changed to 1. On day 2, the price changed to 3."

The prompt provides key details, including the bidder's private value, the total number of bidding days, the current day, the bidding order, previous bids, previous proxy bids, and the current price.

```
You are Bidder Andy.
You are bidding with Bidder Betty, Bidder Charles. {goal}

{RULE EXPLANATION}

Your private value for this item is ${private_value}.
This is the maximum amount you are
willing to pay. Keep this value private.
There are in total {total_periods} days
of bidding and this is day {current_period}.
{ordering}.

Your previous bids are {previous_bid}.
If you have already placed bids, you can only increase your bid
or hold your current bid.
The previous proxy bids are: {transcript}.

The current price is {current_price}.
{Asking Prompt}
```

The decision-making process is again guided by a structured asking prompt that enforces bidding rules and strategic reflection.

```
Your response must use these EXACT
tags below. Don't miss the amount.
```
<PLAN>[ Write your plans for what
bidding strategies to test next.
Be detailed and precise but keep
things succinct and don't repeat yourself.
LIMIT your plan to 50 words. ] </PLAN>
<CHECK> your last bid is
{{last_bid_amount}}, you cannot bid
lower that this value </CHECK>
<ACTION> BID or HOLD  </ACTION>
<AMOUNT> if BID, enter a number here,
e.g. 1, 44. If HOLD, enter 0 </AMOUNT>
```
```

## C  THINKING-MODEL BEHAVES LIKE AN AUCTION THEORIST

As discussed in Section 2.1, our primary methodology avoids RL-post-trained models, like the GPT-o3. This is because thinking-models act as a game theory expert, identifying the underlying structure of the problem and directly computing the optimal solution.

The following examples illustrate this distinction for two common auction types, using a hypothetical private value of $73.

### C.1  EXAMPLE 1: FIRST-PRICE SEALED-BID AUCTION

In a standard three-bidder, first-price sealed-bid auction with private values uniformly distributed from $0 to $99, the model was prompted to determine its bid. Rather than generating a bid based on heuristics or simulated intuition, the model correctly identified and applied the Bayes-Nash Equilibrium (BNE) strategy:

> *For the first round with a valuation of 73, I can apply the symmetric Bayes-Nash equilibrium formula. It's a first-price auction with 3 bidders and uniform valuations from 0 to 99. The bid is roughly 2/3 of 73, which rounds to 49. So, I'll go with a bid of 49.*

### C.2 EXAMPLE 2: SECOND-PRICE SEALED-BID AUCTION

When presented with a second-price sealed-bid auction (also known as a Vickrey auction) with the same parameters, the model again defaulted to an expert analysis. It immediately identified the dominant strategy for this auction type:

> *The scenario gives three bidders, with uniform private values between 0 and 99 in a second-price auction. The strategy is to bid truthfully because it's a Vickrey auction. I should bid 73, matching the value, and avoid any distractions like testing with increments. The answer is clear: 73.*

These responses, while correct from a game-theoretic perspective, do not represent the type of boundedly rational or heuristic-driven behavior that is often observed in human participants and which our main study aims to simulate. This clear divergence in behavior validates our methodological choice to use direct prompting over more complex, reasoning-eliciting approaches.

## D ONE-ROUND SEALED-BID AUCTIONS AND THE DISAPPEARANCE OF OVERBIDDING

A key puzzle in our multi-round settings was: Why do some LLM bidders bid above their assigned value? From the plans and reflections provided by the LLMs, we found two main motivations. First, in multi-round games, the LLMs sometimes purposely inject random or high bids in order to learn how their opponents play or to appear "unpredictable." For example, one LLM bidder declared:

> *I'll adopt a balanced strategy, bidding 65% of my value. I'll also introduce random bids occasionally to disrupt predictability. Monitoring competitor's bids remains essential to adjust my strategy accordingly.*

Second, overbidding in Second Price Sealed-Bid (SPSB) auctions is a well-known "failure mode" even among humans. Although bidding one's true value is a dominant strategy in SPSB, substantial experimental evidence finds that people systematically overbid. Kagel and Levin (1993) report that 67.2% of participants overbid, and even experienced bidders studied by Garratt, Walker, and Wooders (2012) overbid 37.5% of the time. In fact, the persistent puzzle of overbidding in SPSB helped motivate subsequent theoretical work (e.g., Shengwu Li's paper on obviously strategy-proof mechanisms).

To test whether multi-round considerations (information gathering, unpredictability, or learning) were driving the overbidding phenomenon in our LLM bidders, we re-ran all sealed-bid auctions in a strictly single-round setting. We removed any mention that the game might continue beyond one period, and we explicitly stated *"Your top priority is to place a bid that maximizes your expected profit."*

Fig 4 shows the resulting one-round bids across four private-value sealed-bid auction formats. Remarkably, the overbidding phenomenon essentially disappears in one-round interactions.

For First-Price Auction, the bids lie below the diagonal line for most values, much like the Bayesian Nash equilibrium prediction. For SPSB, bids track closely with true values—aligning better with the truthful dominant strategy than in multi-round sessions. For TPSB, the LLM bidders never bid above their value; many actually shade their bids down quite aggressively, often below the theoretical equilibrium curve. For All-Pay Auction, overbidding is also absent here, though some moderate upward bidding at higher valuations remains.

Taken together, these results indicate that LLMs' attempts to be strategic over over repeated interactions is a key factor for overbidding observed in our experiments. When presented with only one shot and explicitly instructed to maximize profit, LLM bidders' tendency to bid above their assigned value virtually vanishes.

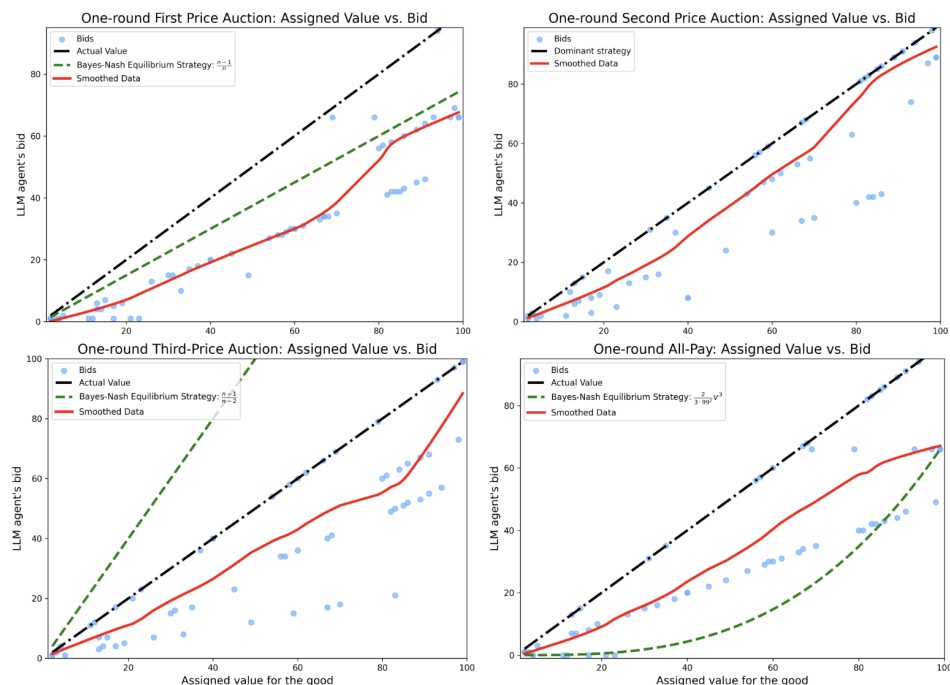

Figure 4: One-round Sealed-bid auction under IPV

# E ROBUSTNESS CHECKS

## E.1 VARYING THE NUMBER OF AGENT IN FIRST-PRICE SEALED-BID AUCTIONS

In FPSB auctions, the theoretically optimal bidding strategy varies based on the number of agents participating. As the number of agents increases, we would expect individual bidders to reduce the extent of bid shading.

To examine this effect, we conducted experiments with FPSB auctions involving 4 and 5 agents, as shown in Fig 5. As the number of agents increases, the Loess-smoothed data curve remains consistently higher than, or approximately aligned with, the predictions of the Bayes-Nash equilibrium. Moreover, we observed variations below the theoretical optimum, along with a few data points exceeding the predicted values. These findings echoes with existing empirical works for first-price auctions (Cox et al., 1988), suggesting that the observed patterns persist across different settings.

## E.2 VARYING CURRENCY IN SPSB AND FPSB AUCTIONS

Additionally, we also tested the prompts in different currencies (specifically, the Euro, the Ruble, the Yen and the Rupee) with the results presented in Figure 6. As is evident from the plots for both the SPSB and FPSB auctions, the key features of the FPSB and SPSB simulation results reported in the main text remain consistent and unchanged in these variations.

## E.3 VARYING LANGUAGE IN RULE EXPLANATIONS IN SPSB AND FPSB

We also tested our existing prompts using different languages (specifically, Spanish, Chinese, Russian and Hindi) with the results presented in Figure 7. It is interesting that the Russian version of FPSB appears to be a stark outlier, with the LLM agent bidding close to the true value even in the first-price auction. For other languages, bids in the second-price auction are generally higher than those in the first-price auction and results qualitatively unchanged from those reported in main body of paper.

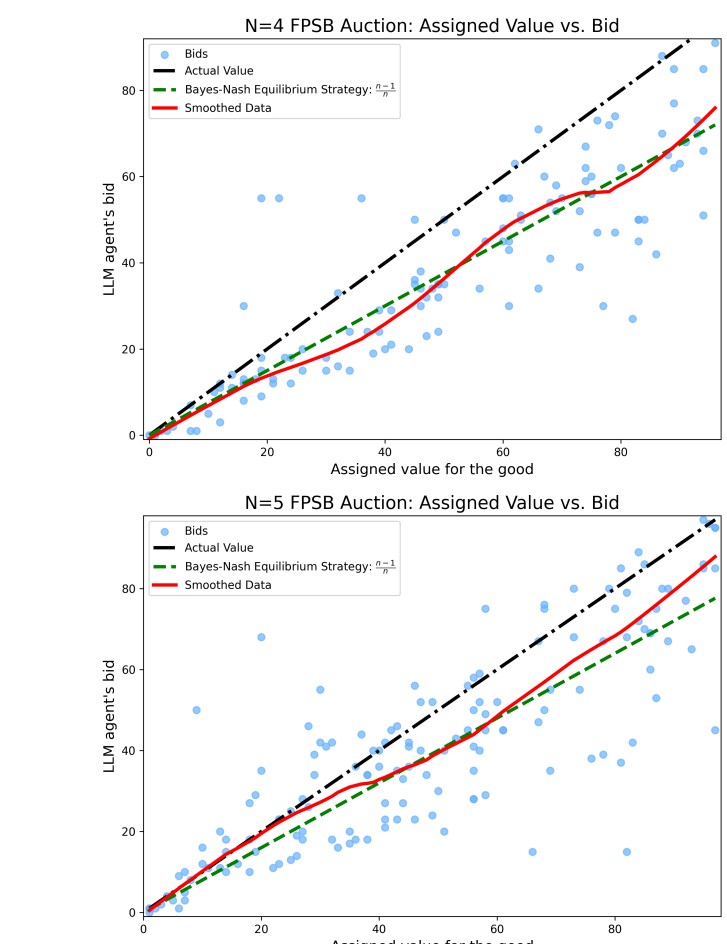

Figure 5: Robustness check of Number of agent in FPSB (IPV setting)

# F  AN ABLATION STUDY WITH AN 'OUT-OF-THE-BOX' LLM AGENT

Throughout Section 3.1, we endowed agents with the ability to plan and reflect as they played the FPSB and SPSB auctions. This 'plan-bid-reflect' loop reflects a modest instantiation of the chain-of-thought paradigm common in the machine learning literature and previous LLM auction experiments (Wei et al., 2022; Fish et al., 2024). However, one might reasonably ask how much of the sophisticated play we observe is actually due to this ability to plan and reflect.

Hence, we also report results of the experiments of Section 3.1 with 'Out-of-the-box' agents – namely, we directly query LLM agents (after providing them with auction rules and their value, as per our simulation process) *without* eliciting their plans or allowing them to reflect on the results of the previous rounds. These IPV experiment results are reported in Figure 8.

Overall, the bidding behaviors exhibited by the agents remain largely monotonic. However, interesting differences emerged between the plan-bid-reflect agents and the out-of-the-box agents. With the out-of-the-box LLM bidders there is now little difference in the bidding behavior between the First-Price and Second-Price auction, a stark difference from the plan-bid-reflect LLM bidders. Now, and unlike in human-subject experiments, no overbidding beyond the agent's value was observed in SPSB auctions. Specifically, in the left panel of Figure 8, no points lie above the bid=value line.

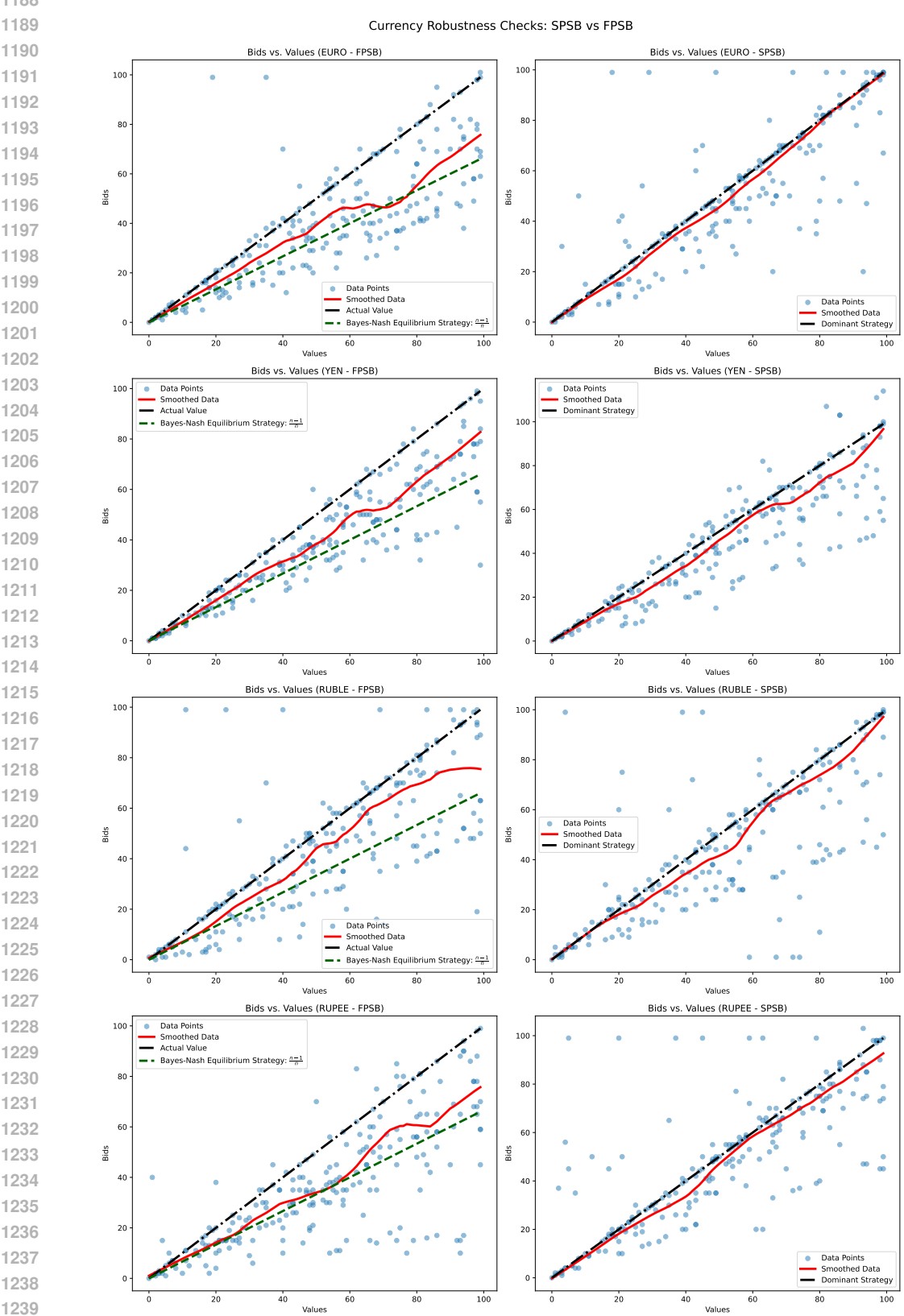

Figure 6: Robustness check with Currency Variation under IPV setting.

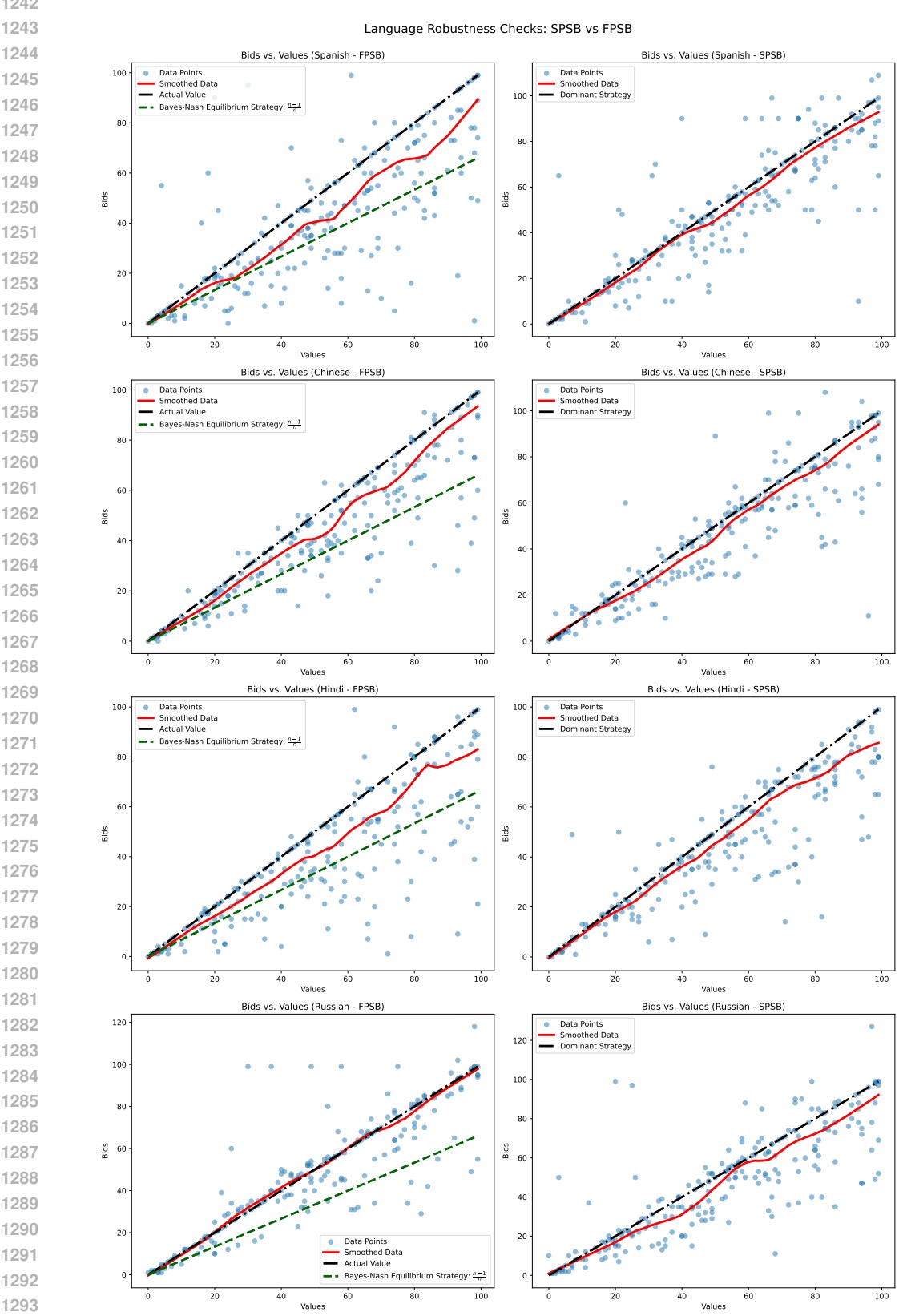

Figure 7: Robustness check with Language Variation (under IPV setting)

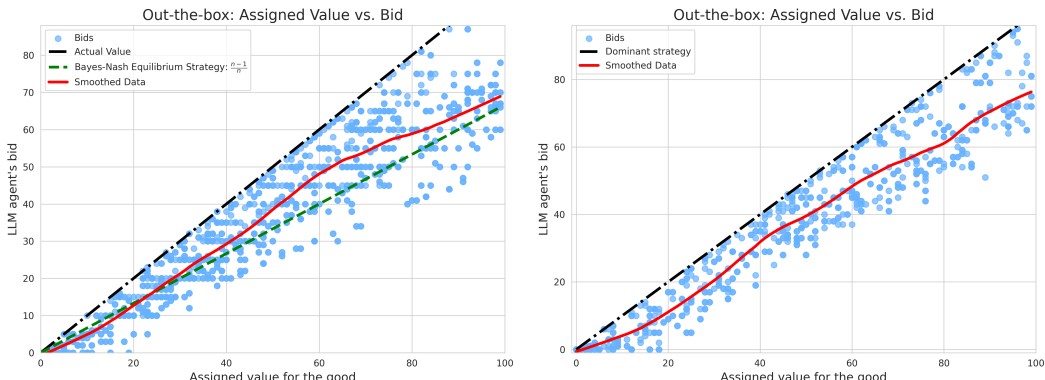

Figure 8: Out-of-the-Box LLM agent in FPSB (left) and SPSB (right) auctions under IPV setting. The blue dots represent bids, the black dotted line is where bid = value (the SPSB dominant strategy), and the green line on the right represents the BNE for FPSB auctions.

## G    STATISTICAL TEST FOR OSP

Below is Table 3, which reports statistical tests for the difference between the SPSB, AC-B and AC auctions.

| Comparison | t-statistic | p-value |
|---|---|---|
| AC v.s. AC-B | 0.71 | 0.48 |
| AC v.s. SPSB | -6.28 | 3.21e-09 |
| AC-B v.s. SPSB | -5.78 | 4.09e-08 |

Table 3: **Two-sample t-tests comparing the mean absolute difference between bids and values in the AC, AC-B and SPSB auctions in an APV setting**. Results demonstrate that both clock auctions have lower mean absolute difference between bid and value vs the sealed-bid auction. However, the clock auctions themselves are not significantly different. Bid data for clock auctions is excluding winners.

## H    SUPPLEMENTARY MATERIALS IN EBAY AUCTIONS

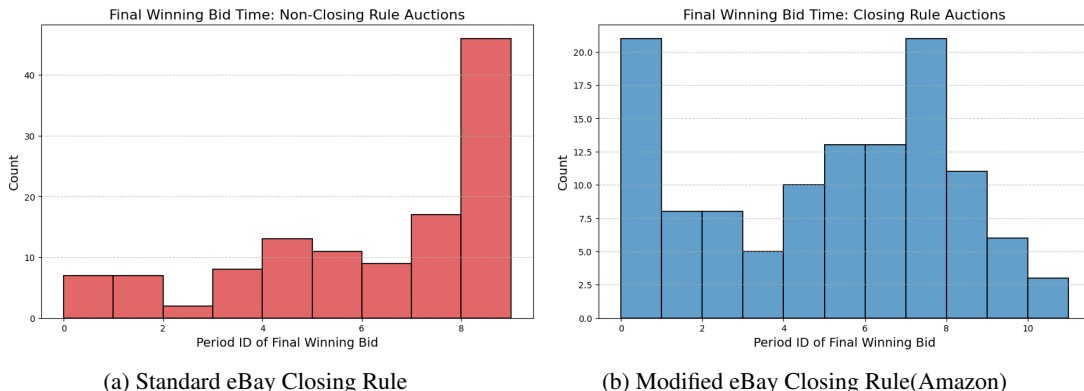

(a) Standard eBay Closing Rule            (b) Modified eBay Closing Rule(Amazon)

Figure 9: **Final Winning Bid Timing in Simulated eBay-style auctions in the IPV environment: standard eBay closing rule vs. modified eBay closing rule.** The modified closing rule introduces the auction-extension logic, moving the timing of the final winning bid substantially earlier in the auction.

| Auction Type 1 | Auction Type 2 | t-statistic | p-value |
|---|---|---|---|
| basic_proxy | closing_rule | 0.020 | 0.984 |
| basic_proxy | hidden_reserve_40 | −0.026 | 0.98 |
| basic_proxy | hidden_reserve_50 | −0.096 | 0.923 |
| basic_proxy | hidden_reserve_60 | −0.071 | 0.944 |
| basic_proxy | closing_rule_hidden_40 | −0.109 | 0.913 |
| basic_proxy | closing_rule_hidden_50 | −0.109 | 0.913 |
| basic_proxy | closing_rule_hidden_60 | −0.119 | 0.905 |
| closing_rule | hidden_reserve_40 | −0.047 | 0.963 |
| closing_rule | hidden_reserve_50 | −0.120 | 0.905 |
| closing_rule | hidden_reserve_60 | −0.093 | 0.926 |
| closing_rule | closing_rule_hidden_40 | −0.133 | 0.895 |
| closing_rule | closing_rule_hidden_50 | −0.133 | 0.895 |
| closing_rule | closing_rule_hidden_60 | −0.144 | 0.886 |
| hidden_reserve_40 | hidden_reserve_50 | −0.071 | 0.944 |
| hidden_reserve_40 | hidden_reserve_60 | −0.045 | 0.964 |
| hidden_reserve_40 | closing_rule_hidden_40 | −0.083 | 0.934 |
| hidden_reserve_40 | closing_rule_hidden_50 | −0.083 | 0.934 |
| hidden_reserve_40 | closing_rule_hidden_60 | −0.093 | 0.927 |
| hidden_reserve_50 | hidden_reserve_60 | 0.026 | 0.98 |
| hidden_reserve_50 | closing_rule_hidden_40 | −0.013 | 0.99 |
| hidden_reserve_50 | closing_rule_hidden_50 | −0.013 | 0.99 |
| hidden_reserve_50 | closing_rule_hidden_60 | −0.020 | 0.984 |
| hidden_reserve_60 | closing_rule_hidden_40 | −0.038 | 0.97 |
| hidden_reserve_60 | closing_rule_hidden_50 | −0.038 | 0.97 |
| hidden_reserve_60 | closing_rule_hidden_60 | −0.046 | 0.963 |
| closing_rule_hidden_40 | closing_rule_hidden_50 | 0.000 | 1 |
| closing_rule_hidden_40 | closing_rule_hidden_60 | −0.006 | 0.995 |
| closing_rule_hidden_50 | closing_rule_hidden_60 | −0.006 | 0.995 |

Table 4: Pairwise t-tests for seller revenue differences across auction types. None of the comparisons show significant revenue differences.

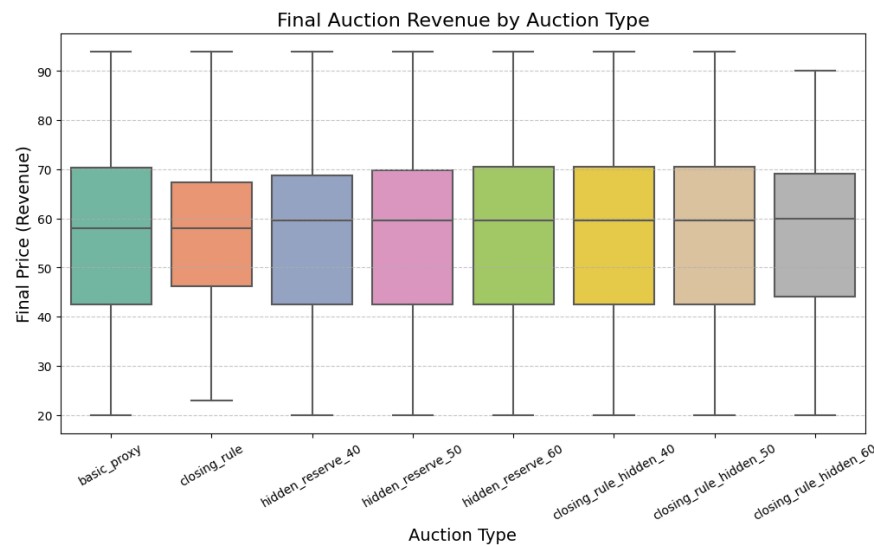

Figure 10: **eBay Revenue by Auction Type, IPV environment** No significant different between any of the settings.

## I RULES EXPLANATION PROMPTS

Below are the rules of each setting for our experiments, which are argument passed to {RULE EXPLANATION} in earlier section.

For the First-Price Sealed-Bid prompt in the IPV setting, the rule is:

```
In this game, you will participate in an auction for a
    prize against {{num_bidders}} other bidders. You
    will play this game for {{n}} rounds.
At the start of each round, bidders will see their
    value for the prize, randomly drawn between \$0
    and \${{private}}, with all values equally likely.
After learning your value, you will submit a bid
    privately at the same time as the other bidders.
    Bids must be between \$0 and \${{private}} in \${{
    increment}} increments.
The highest bidder wins the prize and pays their bid
    amount. If you win, your earnings will increase by
    your value for the prize, and decrease by your
    bid. If you don't win, your earnings will remain
    unchanged.
After each auction, we will display all bids and the
    winner's profits. Ties for the highest bid will be
    resolved randomly.
```

For the Second-Price Sealed-Bid prompt in the IPV setting, the rule is:

```
In this game, you will participate in an auction for a
    prize against {{num_bidders}} other bidders. You
    will play this game for {{n}} rounds.
At the start of each round, bidders will see their
    value for the prize, randomly drawn between $0 and
    ${{private}}, with all values equally likely.
After learning your value, you will submit a bid
    privately at the same time as the other bidders.
```

```
                 Bids must be between $0 and ${{private}} in ${{
                     increment}} increments.
             The highest bidder wins the prize and pays the second-
                 highest bid. If you win, your earnings will
                 increase by your value for the prize, and decrease
                  by the second-highest bid. If you don't win, your
                  earnings will remain unchanged.
             After each auction, we will display all bids and the
                 winner's profits. Ties for the highest bid will be
                  resolved randomly.
```

For the Third-Price Sealed-Bid prompt in the IPV setting, the rule is:

```
             In this game, you will participate in an auction for a
                  prize against {{num_bidders}} other bidders. You
                 will play this game for {{n}} rounds.
             At the start of each round, bidders will see their
                 value for the prize, randomly drawn between $0 and
                  ${{private}}, with all values equally likely.
             After learning your value, you will submit a bid
                 privately at the same time as the other bidders.
                 Bids must be between $0 and ${{private}} in ${{
                 increment}} increments.
             The highest bidder wins the prize and pays the third-
                 highest bid. If you win, your earnings will
                 increase by your value for the prize, and decrease
                  by the third-highest bid. If you don't win, your
                 earnings will remain unchanged. If there are less
                 than three bids, no one will win the auction.
             After each auction, we will display all bids and the
                 winner's profits. Ties for the highest bid will be
                  resolved randomly.
```

For the All-Pay Sealed-Bid prompt in the IPV setting, the rule is:

```
             In this game, you will participate in an auction for a
                  prize against {{num_bidders}} other bidders. You
                 will play this game for {{n}} rounds.
             At the start of each round, bidders will see their
                 value for the prize, randomly drawn between $0 and
                  ${{private}}, with all values equally likely.
             After learning your value, you will submit a bid
                 privately at the same time as the other bidders.
                 Bids must be between $0 and ${{private}} in ${{
                 increment}} increments.
             The highest bidder wins the prize. All bidders (
                 including the winner) pay their submitted bid. If
                 you win, your earnings will increase by your value
                  for the prize, and decrease by your bid. If you
                 don't win, your earnings will still decrease by
                 your bid.
             After each auction, we will display all bids and all
                 bidders' profits. Ties for the highest bid will be
                  resolved randomly.
```

For the Ascending Clock auction (AC) in the APV setting, the rule is

```
In this game, you will participate in an auction for a
    prize against {{num_bidders}} other bidders. You
    will play this game for {{n}} rounds.
At the start of each round, bidders will see their
    value for the prize. Your value for the prize will
    be calculated as follows:
1. First we will randomly draw a common value between
    {{common_low}} and {{common_high}}, with all
    values equally likely.
2. Then, for each bidder, a private taste adjustment
    will be drawn between 0 and {{private}}, with all
    values equally likely.
Your value for the prize is equal to the common value
    plus your private taste adjustment. You will not
    learn the common value or your private taste
    adjustment separately. This means that each person
    in your group may have a different value for the
    prize. However, if you have a high value, it is
    more likely that other people in your group have a
    high value.
The auction proceeds as follows: First, you will learn
    your value for the prize. Then, the auction will
    start. We will display a price to everyone in your
    group that starts at 0 and counts upwards in {{
    increment}} USD increments, up to a maximum of {{
    common_high + private}}. At any point, you can
    choose to leave the auction, and anytime a bidder
    leaves, we will broadcast that information to all
    the remaining bidders.
When there is only one bidder left in the auction,
    that bidder will win the prize at the current
    price. If you win, your earnings will increase by
    your value for the prize, and decrease by the
    current price. If you don't win, your earnings
    will remain unchanged.
After each auction, we will display all bids and the
    winner's profits. Ties for the highest bid will be
    resolved randomly.
```

For the Ascending Clock auction without dropping information (AC-B) in the APV setting, the rule is

```
In this game, you will participate in an auction for a
    prize against {{num_bidders}} other bidders. You
    will play this game for {{n}} rounds.
At the start of each round, bidders will see their
    value for the prize. Your value for the prize will
    be calculated as follows:
1. First we will randomly draw a common value between
    {{common_low}} and {{common_high}}, with all
    values equally likely.
2. Then, for each bidder, a private taste adjustment
    will be drawn between 0 and {{private}}, with all
    values equally likely.
Your value for the prize is equal to the common value
    plus your private taste adjustment. You will not
    learn the common value or your private taste
    adjustment separately. This means that each person
    in your group may have a different value for the
```

```
        prize. However, if you have a high value, it is
        more likely that other people in your group have a
         high value.
    The auction proceeds as follows: First, you will learn
         your value for the prize. Then, the auction will
        start. We will display a price to everyone in your
         group that starts at 0 and counts upwards in {{
        increment}} USD increments, up to a maximum of {{
        common_high + private}}. At any point, you can
        choose to leave the auction, but we will not tell
        any bidder when someone leaves.
    When there is only one bidder left in the auction,
        that bidder will win the prize at the current
        price. If you win, your earnings will increase by
        your value for the prize, and decrease by the
        current price. If you don't win, your earnings
        will remain unchanged.
    After each auction, we will display all bids and the
        winner's profits. Ties for the highest bid will be
         resolved randomly.
```

For the Second-Price Sealed-Bid Auction in the APV setting, the rule is

```
    In this game, you will participate in an auction for a
         prize against {{num_bidders}} other bidders. You
        will play this game for {{n}} rounds.
    At the start of each round, bidders will see their
        value for the prize. Your value for the prize will
         be calculated as follows:
    1. First we will randomly draw a common value between
        {{common_low}} and {{common_high}}, with all
        values equally likely.
    2. Then, for each bidder, a private taste adjustment
        will be drawn between 0 and {{private}}, with all
        values equally likely.
    Your value for the prize is equal to the common value
        plus your private taste adjustment. You will not
        learn the common value or your private taste
        adjustment separately. This means that each person
         in your group may have a different value for the
        prize. However, if you have a high value, it is
        more likely that other people in your group have a
         high value.
    After learning your value, you will submit a bid
        privately at the same time as the other bidders.
        Bids must be between $0 and ${{common_high +
        private}} in ${{increment}} increments.
    The highest bidder wins the prize and pays the second-
        highest bid. If you win, your earnings will
        increase by your value for the prize, and decrease
         by the second-highest bid. If you don't win, your
         earnings will remain unchanged.
    After each auction, we will display all bids and the
        winner's profits. Ties for the highest bid will be
         resolved randomly.
```

For the Second-Price Sealed-Bid auction in the Common Value setting, the rule is

```
In this game, you will participate in an auction for a
    prize against {{num_bidders}} other bidders. You
    will play this game for {{n}} rounds.
At the start of each round, bidders will see their
    perceived value for the prize - a noisy
    measurement of the true value of the prize. Your
    perceived value for the prize will be calculated
    as follows:
1. For each round, a common value will be drawn
    between {{common_low}} and {{common_high}}, with
    all values equally likely to be drawn.
2. For each person, a private noisy adjustment will be
    drawn between -{{private}} and {{private}}, with
    all values equally likely to be drawn.
We will tell you your perceived value, the sum of the
    common value and the private noise adjustment.
    However, everyone's true value for the prize is
    equal to the shared common value.
After learning your perceived value, you will submit a
    bid privately at the same time as the other
    bidders. Bids must be between $0 and ${{
    common_high + private}} in ${{increment}}
    increments.
The highest bidder wins the prize and pays the second-
    highest bid. If you win, your earnings will
    increase by the true value for the prize, and
    decrease by the second-highest bid. If you don't
    win, your earnings will remain unchanged.
After each auction, we will display all bids and the
    winner's profits. Ties for the highest bid will be
    resolved randomly.
```

For the Standard eBay auction with proxy bidding (T1), the rule is

```
In this game, you will participate in an eBay auction
    for an item against {{num_bidders}} other bidders.
    This auction will last for {{num_rounds}} days.
    All dollar amounts in this game are in US Dollars
    ($).

Item Details:
Item Description: {{item_description}}
Item Condition: {{item_condition}}
Your Private Value: At the start of each round,
    bidders will see their value for the item,
    randomly drawn between $0 and ${{private}}, with
    all values equally likely. After learning your
    value for the item, you will submit a maximum bid.
    Bids must be between $0 and ${{private}} in ${{
    increment}} increments.

Auction Format:
This is an eBay auction. The auction starts at ${{
    start_price}} and will last for {{num_rounds}}
    days. eBay uses "proxy bidding." This means that
    if you wish to enter the auction, you should
    submit your maximum bid, and eBay will
    automatically bid on your behalf, up to your
```

```
        maximum, only as much as necessary to maintain
        your position as the highest bidder. Each day you
        will see the current price and have the
        opportunity to increase your maximum bid. If you
        do not want to increase your maximum bid, then
        output HOLD.

    You may place bids at any point during the auction,
        even on the final day. No bidder will know if they
         (or anyone else) is the last bidder on the final
        day.

    The highest bidder wins the prize and pays the auction
         price at the time the auction's clock expired. If
         you win, your earnings will increase by your
        value for the prize, and decrease by the final
        auction price. If you don't win, your earnings
        will remain unchanged. Ties for the highest bid
        will be resolved randomly.
```

For the eBay auction with a modified closing rule (T2), the rule is

```
    In this game, you will participate in an eBay auction
        for an item against {{num_bidders}} other bidders.
         This auction will last for {{num_rounds}} days.
        All dollar amounts in this game are in US Dollars
        ($).

    Item Details:
    Item Description: {{item_description}}
    Item Condition: {{item_condition}}
    Your Private Value: At the start of each round,
        bidders will see their value for the item,
        randomly drawn between $0 and ${{private}}, with
        all values equally likely. After learning your
        value for the item, you will submit a maximum bid.
         Bids must be between $0 and ${{private}} in ${{
        increment}} increments.

    Auction Format:
    This is an eBay auction with a closing rule. The
        auction starts at ${{start_price}} and will last
        for {{num_rounds}} days. eBay uses "proxy bidding
        ." This means that if you wish to enter the
        auction, you should submit your maximum bid, and
        eBay will automatically bid on your behalf, up to
        your maximum, only as much as necessary to
        maintain your position as the highest bidder. Each
         day you will see the current price and have the
        opportunity to increase your maximum bid. If you
        do not want to increase your maximum bid, then
        output HOLD.

    This auction also has a closing rule. This means that
        if a new maximum bid is placed on the last day,
        the auction will automatically extend by another
        day. This extension will continue to be applied as
         long as new maximum bids are placed on the final
```

```
                        day. No bidder will know if they (or anyone else)
                        is the final bidder on the last day.

                    The highest bidder wins the prize and pays the auction
                        price at the time the auction's clock expired. If
                        you win, your earnings will increase by your
                        value for the prize, and decrease by the final
                        auction price. If you don't win, your earnings
                        will remain unchanged. Ties for the highest bid
                        will be resolved randomly.
```

For the standard eBay auction with a hidden reserve price (T3), the rule is

```
                    In this game, you will participate in an eBay auction
                        for an item against {{num_bidders}} other bidders.
                        This auction will last for {{num_rounds}} days.
                        All dollar amounts in this game are in US Dollars
                        ($).

                    Item Details:
                    Item Description: {{item_description}}
                    Item Condition: {{item_condition}}
                    Your Private Value: At the start of each round,
                        bidders will see their value for the item,
                        randomly drawn between $0 and ${{private}}, with
                        all values equally likely. After learning your
                        value for the item, you will submit a maximum bid.
                        Bids must be between $0 and ${{private}} in ${{
                        increment}} increments.

                    Auction Format:
                    This is an eBay auction with a hidden reserve price.
                        The auction starts at ${{start_price}} and will
                        last for {{num_rounds}} days. eBay uses "proxy
                        bidding." This means that if you wish to enter the
                        auction, you should submit your maximum bid, and
                        eBay will automatically bid on your behalf, up to
                        your maximum, only as much as necessary to
                        maintain your position as the highest bidder. Each
                        day you will see the current price and have the
                        opportunity to increase your maximum bid. If you
                        do not want to increase your maximum bid, then
                        output HOLD.

                    You may place bids at any point during the auction,
                        even on the final day. However, if no bidder
                        produces a maximum bid above the hidden reserve
                        price, the seller will retain the good and the
                        bidders will be informed that there is currently
                        no bidder in the lead. No bidder will know if they
                        (or anyone else) is the final bidder on the last
                        day.

                    If the reserve is met, the highest bidder wins the
                        prize and pays the auction price at the time the
                        auction's clock expired. If you win, your earnings
                        will increase by your value for the prize, and
                        decrease by the final auction price. If you don't
```

win, your earnings will remain unchanged. Ties for
    the highest bid will be resolved randomly.

For the eBay auction with a modified closing rule and a hidden reserve price (T4), the rule is

    In this game, you will participate in an eBay auction
        for an item against {{num_bidders}} other bidders.
         This auction will last for {{num_rounds}} days.
        All dollar amounts in this game are in US Dollars
        ($).

    Item Details:
    Item Description: {{item_description}}
    Item Condition: {{item_condition}}
    Your Private Value: At the start of each round,
        bidders will see their value for the item,
        randomly drawn between $0 and ${{private}}, with
        all values equally likely. After learning your
        value for the item, you will submit a maximum bid.
         Bids must be between $0 and ${{private}} in ${{
        increment}} increments.

    Auction Format:
    This is an eBay auction with a hidden reserve price
        and a closing rule. The auction starts at ${{
        start_price}} and will last for {{num_rounds}}
        days. eBay uses "proxy bidding." This means that
        if you wish to enter the auction, you should
        submit your maximum bid, and eBay will
        automatically bid on your behalf, up to your
        maximum, only as much as necessary to maintain
        your position as the highest bidder. Each day you
        will see the current price and have the
        opportunity to increase your maximum bid. If you
        do not want to increase your maximum bid, then
        output HOLD.

    You may place bids at any point during the auction,
        even on the final day. However, if no bidder
        produces a maximum bid above the hidden reserve
        price, the seller will retain the good and the
        bidders will be informed that there is currently
        no bidder in the lead.

    This auction also has a closing rule. This means that
        if a new maximum bid is placed on the last day,
        the auction will automatically extend by another
        day. This extension will continue to be applied as
         long as new maximum bids are placed on the final
        day. No bidder will know if they (or anyone else)
        is the last bidder on the final day.

    If the reserve is met, the highest bidder wins the
        prize and pays the auction price at the time the
        auction's clock expired. If you win, your earnings
         will increase by your value for the prize, and
        decrease by the final auction price. If you don't
        win, your earnings will remain unchanged. Ties for
         the highest bid will be resolved randomly.

For the Menu-Description intervention, the rule is

```
In this game, you will participate in an auction for a
    prize against {{num_bidders}} other bidders. You
    will play this game for {{n}} rounds.
At the start of each round, bidders will see their
    value for the prize, randomly drawn between $0 and
    ${{private}}, with all values equally likely.
After learning your value, you will submit a bid
    privately at the same time as the other bidders.
    Bids must be between $0 and ${{private}} in ${{
    increment}} increments.
Your "price to win" the item will be set to the
    highest bid placed by any other player. If your
    bid is higher than this "price to win," then you
    will win the item and pay this price. If you don't
    win, your earnings will remain unchanged.
After each auction, we will display all bids and the
    winner's profits. Ties for the highest bid will be
    resolved randomly.
```

For the Clock-Description intervention, the rule is

```
In this game, you will participate in an auction for a
    prize against {{num_bidders}} other bidders. You
    will play this game for {{n}} rounds.
    At the start of each round, bidders will see their
        value for the prize, randomly drawn between
        $0 and ${{private}}, with all values equally
        likely.
    After learning your value, you will submit a bid
        privately at the same time as the other
        bidders. Bids must be between $0 and ${{
        private}} in ${{increment}} increments.
**FIRST STAGE: Sealed Bid**
    You will submit a sealed bid privately at the same
        time as the other bidders. This bid will
        serve as your automatic exit price in the next
        stage.
**SECOND STAGE: Ascending Clock (Simulation)**
    After the sealed bid stage, we will simulate an
        ascending clock auction.
    The clock will start at $0 and increase in
        increments of ${{increment}}.
    The clock will display the current price. You will
        also see that there are a total of {{num
        bidders}} bidders participating, although you
        do not know other bidder's values.
    If the current price on the clock reaches or
        exceeds your sealed bid from the first stage,
        you will automatically exit the auction. The
        auction ends when only one bidder is left
        remaining in the second stage based on their
        bid from the first stage.
**END OF AUCTION**
If you win, your earnings will increase by your value
    for the prize and decrease by the clock price at
    the end of the auction. If you don't win, your
    earnings will remain unchanged.
```

```
                    After each auction, we will display all bids and
                        the winner's profits. Ties for the highest bid
                         will be resolved randomly.
```

For the Nash-Deviation intervention, the rule is

```
        In this game, you will participate in an auction for a
            prize against {{num_bidders}} other bidders. You
            will play this game for {{n}} rounds.
        At the start of each round, bidders will see their
            value for the prize, randomly drawn between $0 and
            ${{private}}, with all values equally likely.
        After learning your value, you will submit a bid
            privately at the same time as the other bidders.
            Bids must be between $0 and ${{private}} in ${{
            increment}} increments.
        The highest bidder wins the prize and pays the second-
            highest bid. If you win, your earnings will
            increase by your value for the prize, and decrease
            by the second-highest bid. If you don't win, your
            earnings will remain unchanged.
        First, come up with a possible bid given your value.
            Then, think through your bidding strategy step by
            step. How do you expect others to bid? If others
            bid like this and you bid down, what happens? If
            others bid like this and you bid up, what happens?
        Think through all the ways you could deviate from your
            current bidding strategy, and settle on the best
            possible strategy. Then return your bid.
        After each auction, we will display all bids and the
            winner's profits. Ties for the highest bid will be
            resolved randomly.
```

For the Direct-Revelation intervention, the rule is

```
        In this game, you will participate in an auction for a
            prize against {{num_bidders}} other bidders. You
            will play this game for {{n}} rounds.
        At the start of each round, bidders will see their
            value for the prize, randomly drawn between $0 and
            ${{private}}, with all values equally likely.
        After learning your value, you will submit a bid
            privately at the same time as the other bidders.
            Bids must be between $0 and ${{private}} in ${{
            increment}} increments.
        The highest bidder wins the prize and pays the second-
            highest bid. If you win, your earnings will
            increase by the value for the prize, and decrease
            by the second-highest bid. If you don't win, your
            earnings will remain unchanged.
        Economists have studied this game and they've found
            that the dominant strategy - always the right
            thing to do whatever the bids of others - of this
            game is to bid your value. Consider if they might
            be right, and then generate your bidding strategy.
        After each auction, we will display all bids and the
            winner's profits. Ties for the highest bid will be
            resolved randomly.
```

For the Wrong Direct Revelation intervention, the rule is

```
In this game, you will participate in an auction for a
    prize against {{num_bidders}} other bidders. You
    will play this game for {{n}} rounds.
At the start of each round, bidders will see their
    value for the prize, randomly drawn between $0 and
    ${{private}}, with all values equally likely.
After learning your value, you will submit a bid
    privately at the same time as the other bidders.
    Bids must be between $0 and ${{private}} in ${{
    increment}} increments.
The highest bidder wins the prize and pays the second-
    highest bid. If you win, your earnings will
    increase by the value for the prize, and decrease
    by the second-highest bid. If you don't win, your
    earnings will remain unchanged.
Economists have studied this game and theyve found
    that the dominant strategy - always the right
    thing to do whatever the bids of others - of this
    game is to bid 50% your value. Consider if they
    might be right, and then generate your bidding
    strategy.
After each auction, we will display all bids and the
    winner's profits. Ties for the highest bid will be
    resolved randomly.
```

Below are the prompts used in the robustness check with Currency and Language:

For the FPSB auction with another currency, the rule is

```
In this game, you will participate in an auction for a
    prize against {{num_bidders}} other bidders.
At the start of each round, bidders will see their
    value for the prize, randomly drawn between {{
    currency symbol}} 0 and {{currency symbol}}{{
    private}}, with all values equally likely.
After learning your value, you will submit a bid
    privately at the same time as the other bidders.
    Bids must be between {{currency symbol}}0 and {{
    currency symbol}}{{private}} in {{currency symbol
    }}{{increment}} increments.
The highest bidder wins the prize and pays their bid
    amount. This means that, if you win, we will add
    to your earnings the value for the prize, and
    subtract from your earnings your bid. If you don't
     win, your earnings remain unchanged.
After each auction, we will display all bids and
    profits. Ties for the highest bid will be resolved
     randomly.
```

For the SPSB auction with Euro currency, the rule is

```
In this game, you will participate in an auction for a
    prize against {{num_bidders}} other bidders.
At the start of each round, bidders will see their
    value for the prize, randomly drawn between {{
    currency symbol}}0 and {{currency symbol}}{{
    private}}, with all values equally likely.
```

```
        After learning your value, you will submit a bid
            privately at the same time as the other bidders.
            Bids must be between {{currency symbol}}0 and {{
            currency symbol}}{{private}} in {{currency symbol
            }}{{increment}} increments.
        The highest bidder wins the prize and pays the second-
            highest bid. This means that, if you win, we will
            add to your earnings the value for the prize, and
            subtract from your earnings the second-highest bid
            . If you don't win, your earnings remain unchanged
            .
        After each auction, we will display all bids and the
            winner's profits. Ties for the highest bid will be
             resolved randomly.
```

For all the above, the currency symbol will be replaced with the respective currency symbol of Euro, Yen, Rupee, or Ruble as in the experiments.

For the FPSB auction with Spanish, the rule is

En este juego, participarás en una subasta por un premio contra {{num_bidders}} otros postores.

Al inicio de cada ronda, los postores verán su valor por el premio, extraído aleatoriamente entre $0 y ${{private}}, con todos los valores igualmente probables.

Después de conocer tu valor, presentarás una oferta de forma privada al mismo tiempo que los demás postores. Las ofertas deben estar entre $0 y ${{private}} en incrementos de ${{increment}}.

El postor más alto gana el premio y paga el monto de su oferta. Esto significa que, si ganas, agregaremos a tus ganancias el valor del premio, y restaremos de tus ganancias tu oferta. Si no ganas, tus ganancias permanecerán sin cambios.

Después de cada subasta, mostraremos todas las ofertas y ganancias. Los empates por la oferta más alta se resolverán aleatoriamente.

For the SPSB auction with Spanish, the rule is

En este juego, participarás en una subasta por un premio contra {{num_bidders}} otros postores.

Al inicio de cada ronda, los postores verán su valor por el premio, extraído aleatoriamente entre $0 y ${{private}}, con todos los valores igualmente probables.

Después de conocer tu valor, presentarás una oferta de forma privada al mismo tiempo que los demás postores. Las ofertas deben estar entre $0 y ${{private}} en incrementos de ${{increment}}.

El postor más alto gana el premio y paga la segunda oferta más alta. Esto significa que, si ganas, agregaremos a tus ganancias el valor del premio, y restaremos de tus ganancias la segunda oferta más alta. Si no ganas, tus ganancias permanecerán sin cambios.

Después de cada subasta, mostraremos todas las ofertas y las ganancias del ganador. Los empates por la oferta más alta se resolverán aleatoriamente.

For the FPSB auction with Chinese, the rule is

```
在这个游戏中，你将参与一个与{{num_bidders}}其他竞标者竞争奖品的拍卖。
在每一轮开始时，竞标者将看到他们对奖品的价值，该价值在 0 美元到 ¥{{private}} 美元之间随机抽取，所有数值的可能性均相等。
在了解你的价值后，你将与其他竞标者同时私下提交一个投标。投标必须在 0 美元到 ¥{{private}} 美元之间，以 ¥{{increment}} 美元的增量递增。
出价最高者赢得奖品并支付第二高出价。这意味着，如果你赢了，我们将把奖品的价值添加到你的收入中，并从你的收入中减去第二高出价。如果你没有赢，你的收入将保持不变。
每次拍卖后，我们将显示所有投标和获胜者的利润。最高出价的平局将随机解决。
```

For the SPSB auction with Chinese, the rule is

```
在这个游戏中，你将参与一个与{{num_bidders}}其他竞标者竞争奖品的拍卖。
在每一轮开始时，竞标者将看到他们对奖品的价值，该价值在 0 美元到 ¥{{private}} 美元之间随机抽取，所有数值的可能性均相等。
在了解你的价值后，你将与其他竞标者同时私下提交一个投标。投标必须在 0 美元到 ¥{{private}} 美元之间，以 ¥{{increment}} 美元的增量递增。
出价最高者赢得奖品并支付其投标金额。这意味着，如果你赢了，我们将把奖品的价值添加到你的收入中，并从你的收入中减去你的投标。如果你没有赢，你的收入将保持不变。
每次拍卖后，我们将显示所有投标和利润。最高出价的平局将随机解决。
```

For the FPSB auction with Hindi, the rule is

```
इस खेल में, आप {{num_bidders}} अन्य बोलीदाताओं के खिलाफ एक पुरस्कार के लिए नीलामी में भाग लेंगे।
प्रत्येक दौर की शुरुआत में, बोलीदाताओं को पुरस्कार के लिए अपना मूल्य दिखाई देगा, जो ₹0 और ₹{{private}} के बीच यादृच्छिक रूप से निकाला गया है, सभी मूल्य समान रूप से संभावित हैं।
अपना मूल्य जानने के बाद, आप अन्य बोलीदाताओं के साथ एक ही समय में निजी तौर पर एक बोली जमा करेंगे। बोलियाँ ₹0 और ₹{{private}} के बीच ₹{{increment}} की वृद्धि में होनी चाहिए।
सबसे अधिक बोली लगाने वाला पुरस्कार जीतता है और अपनी बोली राशि का भुगतान करता है। इसका मतलब है कि, यदि आप जीतते हैं, तो हम आपकी कमाई में पुरस्कार के लिए मूल्य जोड़ेंगे, और आपकी कमाई से आपकी बोली घटाएंगे। यदि आप नहीं जीतते हैं, तो आपकी कमाई अपरिवर्तित रहेगी।
प्रत्येक नीलामी के बाद, हम सभी बोलियों और मुनाफा प्रदर्शित करेंगे। सबसे अधिक बोली के लिए टाई को यादृच्छिक रूप से हल किया जाएगा।
```

For the SPSB auction with Hindi, the rule is

```
इस खेल में, आप {{num_bidders}} अन्य बोलीदाताओं के खिलाफ एक पुरस्कार के लिए नीलामी में भाग लेंगे।
प्रत्येक दौर की शुरुआत में, बोलीदाताओं को पुरस्कार के लिए अपना मूल्य दिखाई देगा, जो ₹0 और ₹{{private}} के बीच यादृच्छिक रूप से निकाला गया है, सभी मूल्य समान रूप से संभावित हैं।
अपना मूल्य जानने के बाद, आप अन्य बोलीदाताओं के साथ एक ही समय में निजी तौर पर एक बोली जमा करेंगे। बोलियाँ ₹0 और ₹{{private}} के बीच ₹{{increment}} की वृद्धि में होनी चाहिए।
सबसे अधिक बोली लगाने वाला पुरस्कार जीतता है और दूसरी सबसे अधिक बोली का भुगतान करता है। इसका मतलब है कि, यदि आप जीतते हैं, तो हम आपकी कमाई में पुरस्कार के लिए मूल्य जोड़ेंगे, और आपकी कमाई से दूसरी सबसे अधिक बोली घटाएंगे। यदि आप नहीं जीतते हैं, तो आपकी कमाई अपरिवर्तित रहेगी।
प्रत्येक नीलामी के बाद, हम सभी बोलियों और विजेता के मुनाफे प्रदर्शित करेंगे। सबसे अधिक बोली के लिए टाई को यादृच्छिक रूप से हल किया जाएगा।
```

For the FPSB auction with Russian, the rule is

```
В этой игре вы примете участие в аукционе за приз против {{num_bidders}} других участников.
В начале каждого раунда участники увидят свою ценность приза, случайно выбранную в диапазоне от 0 до ₽{{private}} долларов, при этом все значения равновероятны.
После того, как вы узнаете свою ценность, вы подадите заявку в частном порядке одновременно с другими участниками. Ставки должны быть в диапазоне от 0 до ₽{{private}} долларов с шагом в ₽{{increment}} долларов.
Участник, предложивший самую высокую цену, выигрывает приз и платит сумму своей ставки. Это означает, что если вы выиграете, мы добавим к вашему заработку стоимость приза и вычтем из вашего заработка вашу ставку. Если вы не выиграете, ваш заработок останется неизменным.
После каждого аукциона мы покажем все ставки и прибыль. Ничьи по самой высокой ставке будут разрешены случайным образом.
```

For the SPSB auction with Russian, the rule is

```
В этой игре вы примете участие в аукционе за приз против {{num_bidders}} других участников.
В начале каждого раунда участники увидят свою ценность приза, случайно выбранную в диапазоне от 0 до ₽{{private}} долларов, при этом все значения равновероятны.
После того, как вы узнаете свою ценность, вы подадите заявку в частном порядке одновременно с другими участниками. Ставки должны быть в диапазоне от 0 до ₽{{private}} долларов с шагом в ₽{{increment}} долларов.
Участник, предложивший самую высокую цену, выигрывает приз и платит вторую по величине ставку. Это означает, что если вы выиграете, мы добавим к вашему заработку стоимость приза и вычтем из вашего заработка вторую по величине ставку. Если вы не выиграете, ваш заработок останется неизменным.
После каждого аукциона мы покажем все ставки и прибыль победителя. Ничьи по самой высокой ставке будут разрешены случайным образом.
```

## J  INTERVENTION SUPPLEMENTARY MATERIALS

First, we report an additional Figure for visualization of the effect of the 6 interventions.

Now, we present the prompts in additional detail:

**Menu-Description**

> Your "price to win" the item will be set to the highest bid placed by any other player. If your bid is higher than this "price to win," then you will win the item and pay this price. If you don't win, your earnings will remain unchanged.

**Clock-Description**

> FIRST STAGE: Sealed Bid You will submit a sealed bid privately at the same time as the other bidders. This bid will serve as your automatic exit price in the next stage.
> SECOND STAGE: Ascending Clock (Simulation)
> After the sealed bid stage, we will simulate an ascending clock auction. The clock will start at 0 and increase in increments of increment. The clock will display the

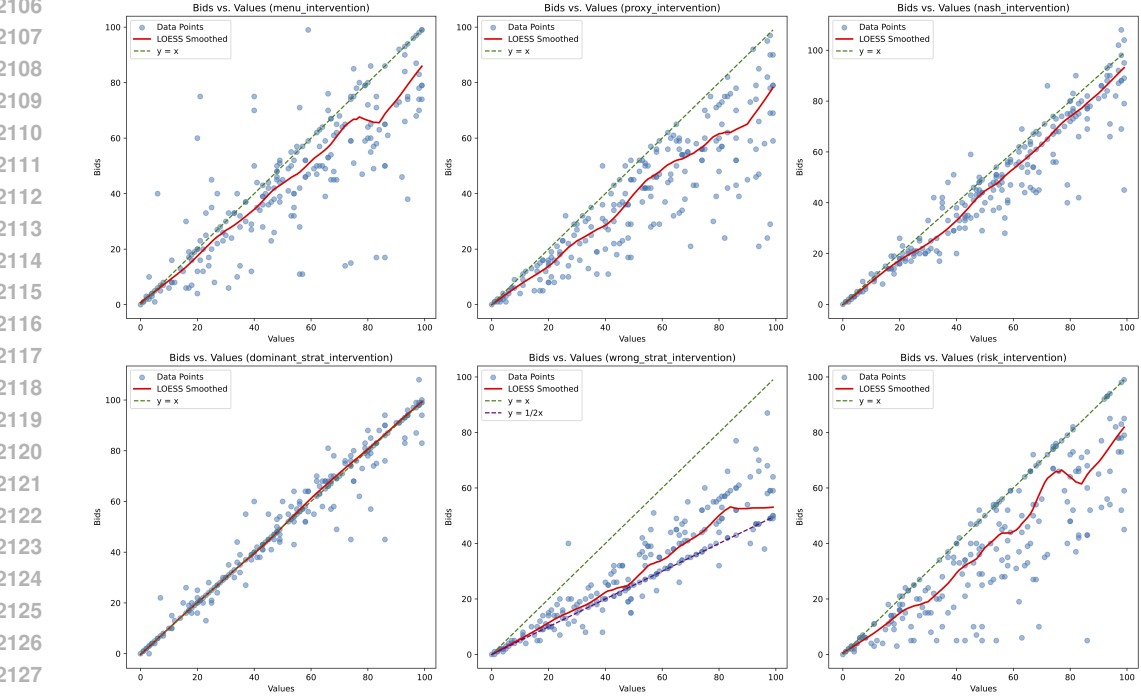

Figure 11: **Results for SPSB with the six interventions, IPV environment.** The green dotted line represents the scenario of bid=value, blue dots are bids from agents, and the red line is the LOESS smoothed curve of bids. Menu and proxy framings reduce scatter but still hug the region below the green $y = x$ line. The Nash-deviation framing tilts the curve almost exactly onto the 45-degree line. The dominant-strategy advice panel shows the tightest clustering, whereas the 'Wrong Strategy' panel visibly drags the red curve toward $y = v/2$.

*current price. You will also see that there are a total of {num_bidders} bidders participating, although you do not know other bidder's values. If the current price on the clock reaches or exceeds your sealed bid from the first stage, you will automatically exit the auction. The auction ends when only one bidder is left remaining in the second stage based on their bid from the first stage.*
*END OF AUCTION*

**Nash Deviation**

*First, come up with a possible bid given your value. Then, think through your bidding strategy step by step. How do you expect others to bid? If others bid like this and you bid down, what happens? If others bid like this and you bid up, what happens? Think through all the ways you could deviate from your current bidding strategy, and settle on the best possible strategy.*

**Direct Revelation**

*Economists have studied this game and they've found that the dominant strategy — always the right thing to do whatever the bids of others — of this game is to bid your value. Consider if they might be right, and then generate your bidding strategy.*

**Wrong Direct Revelation**

*Economists have studied this game and they've found that the dominant strategy — always the right thing to do whatever the bids of others — of this game is to bid 50% your value. Consider if they might be right, and then generate your bidding strategy.*

**Risk Neutrality**

> *You are neutral about risk. This means that you are the type of person that would*
> *pay $5 for a coin toss where you got $0.00 on tails and $10.00 on heads.*

## K    ETHICS STATEMENT

This work proposed a new method to generate synthetic data for designing more efficient auctions. This may lead to more equitable and efficient resource allocation outcomes if there are new design generated through our methodology. Our experiments are purely based on LLM simulation and compare prior lab data, and no new data was collected from human subjects.

## L    REPRODUCIBILITY STATEMENT

The experimental setup, simulator parameters, and all LLM prompts are described in Section 2 and the appendices. The source code to reproduce all results is included in the supplementary material and will be made public, and we included it in the submission.

## M    LLM USAGE

LLMs were used as assistive tools for editing, grammar, and code debugging during the preparation of this manuscript. The core research ideas, theoretical framework, experimental design, and analysis were conceived and executed by the human authors, who reviewed and take full responsibility for all content.

