# OpenReview forum: "Learning from Synthetic Labs: Language Models as Auction Participants"
_ICLR.cc/2026/Conference — ICLR 2026 Conference Withdrawn Submission_

### Official Review · Reviewer_o6Zh · 2025-10-16

**Soundness:** 3
**Presentation:** 3
**Contribution:** 2
**Rating:** 6
**Confidence:** 4

**Summary:**

The paper examines the behavior of LLMs in simulated auctions and the extent to which their behavior aligns with both theoretical models of optimal bidding strategies and reported human behavior in such environments. The paper considers several experimental settings, including first- and second-price sealed-bid auctions, clock auctions, and an eBay-like auction. The paper examines several intervention forms in the auction framing and studies their effect on deviation from truthful bidding in the (truthful) second-price auction.
 The results indicate that many of the behavioral patterns observed in human participants carry over to LLM participants (e.g., risk-aversion, improved rationality in obviously strategyproof auctions, and end-of-auction 'sniping').

**Strengths:**

1. The paper offers a rigorous and timely analysis at the intersection of AI, economic theory, and human behavior. It provides valuable insights into how closely LLMs align with theoretical predictions and experimental findings with human participants, highlighting their potential as tools for behavioral economics simulations. In fact, regardless of the degree of alignment with human behavior, I believe that understanding LLM behavior in complex decision-making is crucial, as such models are increasingly deployed as active participants in real-world systems.

2. The paper provides a rich and comprehensive experimental design with multiple auction formats, intervention types, and robustness checks (language, currency, number of players), which strengthens its contribution to the ML community.

**Weaknesses:**

1. The paper’s literature review is somewhat underdeveloped, making it difficult to situate its contribution within the broader research landscape. Strengthening this section would help clarify its novelty and relevance. In particular, it would benefit from engaging with adjacent lines of work, including:
- studies using LLMs to simulate strategic behavior beyond auction settings [1,2,3];
- research on predicting and interpreting human decision-making with LLMs [4,5];
- alternative approaches to evaluating the rationality of LLMs [6,7].

2. The paper uncovers several interesting behavioral patterns in LLMs, but it does not explore how robust these findings are across prompts, models, and experimental conditions. For example:
- Since the results suggest that LLMs exhibit risk-averse behavior, it would be valuable to test whether these patterns can be influenced through prompting -- either directly (by instructing risk-seeking or risk-averse behavior) or indirectly (by assigning different personas and comparing their risk attitudes).
- Similarly, the observation that LLMs act more rationally in obviously strategyproof auctions raises the question of whether this behavior persists with smaller models or those lacking explicit reasoning capabilities.

(Just to make sure I haven’t overlooked anything — I reviewed Sections E–F for answers to these kinds of questions, but as far as I can tell, those sections focus on other types of robustness checks. Please correct me if I’m mistaken or if I missed any discussion relevant to the points above)

[1] Akata, E., Schulz, L., Coda-Forno, J., Oh, S. J., Bethge, M., & Schulz, E. (2025). Playing repeated games with large language models. Nature Human Behaviour, 1-11.

[2] Shapira, E., Madmon, O., Reinman, I., Amouyal, S. J., Reichart, R., & Tennenholtz, M. (2024). Glee: A unified framework and benchmark for language-based economic environments. arXiv preprint arXiv:2410.05254.

[3] Abdelnabi, S., Gomaa, A., Sivaprasad, S., Schönherr, L., & Fritz, M. (2024). Cooperation, competition, and maliciousness: Llm-stakeholders interactive negotiation. Advances in Neural Information Processing Systems, 37, 83548-83599.

[4] Shapira, E., Madmon, O., Reichart, R., & Tennenholtz, M. (2024). Can llms replace economic choice prediction labs? the case of language-based persuasion games. arXiv preprint arXiv:2401.17435.

[5] Liu, R., Geng, J., Peterson, J. C., Sucholutsky, I., & Griffiths, T. L. (2024). Large language models assume people are more rational than we really are. arXiv preprint arXiv:2406.17055.

[6] Macmillan-Scott, O., & Musolesi, M. (2024). (Ir) rationality and cognitive biases in large language models. Royal Society Open Science, 11(6), 240255.

[7] Alsagheer, D., Karanjai, R., Shi, W., Diallo, N., Lu, Y., Beydoun, S., & Zhang, Q. (2024). Evaluating irrationality in large language models and open research questions. In Proceedings of the HEAL Workshop at CHI. ACM.

**Questions:**

Do you have an idea whether your findings are robust to variations in temperature and/or the underlying value distribution (i.e., beyond uniform distribution)? In particular, I am curious about the case of temperature = 0, as this choice arguably better reflects the model’s inherent behavioral tendencies.

Additionally, I am open to hearing your thoughts or responses to the weaknesses highlighted above.

---

### Official Review · Reviewer_dhAz · 2025-10-21

**Soundness:** 1
**Presentation:** 2
**Contribution:** 1
**Rating:** 0
**Confidence:** 4

**Summary:**

The paper proposes using LLM agents to generate synthetic data for classic auction environments (e.g., FPSB, SPSB, clock/OSP, and an eBay-style setting). The authors claim (1) a flexible, low-cost framework for running many auction instances with LLM “bidders,” (2) qualitative replication of well-known empirical patterns from experimental economics (e.g., behavior under strategy-proof formats and eBay-style sniping), and (3) prompt-level “interventions” that purportedly move agent behavior closer to theoretical predictions. They position the approach as a proof-of-concept toward using LLMs as proxies for human subjects in mechanism design studies.

**Strengths:**

The question of whether LLM agents can substitute for human subjects in economic mechanism experiments is timely and interesting.

**Weaknesses:**

1. Poor clarity and factual accuracy: Important auction acronyms are not clearly defined in the Introduction; figures use very small fonts; and at least one factual claim appears misleading (e.g., attributing inspiration to prior work purportedly using GPT-4o where, to the best of my knowledge, Horton23 used GPT-3).
2. Several headline comparisons rely on different granularity and thresholds than the classic studies (e.g., different numbers of bidders, coarser bid grids, broader “match-to-value” criteria). Conclusions such as “similar performance” are therefore not like-for-like and should be qualified or re-analyzed with matched protocols.
3. The “interventions” often reveal dominant strategies or otherwise prime desired responses, making it difficult to attribute improvements to emergent strategic reasoning rather than compliance with instructions.
4. Most results hinge on a single foundation model and a narrow hyperparameter regime (e.g., temperature). There is little evidence that the findings persist across models, decoding choices, or ablations.

**Questions:**

Can you re-run the principal analyses with exactly matched protocols to the cited human experiments from Kagle93 (same bidder counts, bid grid, payoff reporting, and “match-to-value” thresholds), and report side-by-side metrics?

---

### Official Review · Reviewer_77vE · 2025-11-10

**Soundness:** 3
**Presentation:** 2
**Contribution:** 1
**Rating:** 2
**Confidence:** 4

**Summary:**

The paper demonstrates Language models' ability to mimic human behavior in auction scenarios. They simulate 1,000+ auction experiments and even replicate theoretical results in a quasi-real-world setting.

**Strengths:**

This paper demonstrates at scale how LLMs can mimic human behavior (or approach theoretical results) in thousands of auction scenarios. I appreciate authors looking into LLM simulations, as it has immense potential in simulating real-world experiments, feeding into the direction of AI-driven science. Overall, the paper is well written.

**Weaknesses:**

- I completely fail to understand the technical novelty of this paper. Considering the ICLR audience, I do think the papers need to explicitly demonstrate why the proposed framework works as presented.
- While the finding is exciting (still a bit limited in terms of applicability), it is hard to assess what alternative approach would have worked to achieve similar performance. The work is plagued by a dire lack of reasonable baselines.
- In fact, as far as I remember, LLM's ability to perform well in an auction was shown two years back, with AucArea ("Put Your Money Where Your Mouth Is: Evaluating Strategic Planning and Execution of LLM Agents in an Auction Arena"). The related work mentions this but fails to strike a clear difference with the proposed approach.
- " We see the main contribution of this work as establishing a framework for considering LLM evidence as a proxy for human evidence in mechanism design" -- turning back to the framework, first, what is the framework, rather than scaffolding LLM API calls? That could still be very effective (as shown in this case, similar to many other cases), but does this framework (since so simple) extend to other examples of mechanism design? Or even broader, other strategic situations? Even if it does (which this paper does not provably demonstrate), what do we know about how LLMs would behave in out-of-distribution scenarios, which could be critical to establish fidelity in LLM simulations?
"Can Language Models Serve as Text-Based World Simulators?" (https://arxiv.org/pdf/2406.06485?) raises serious concerns about such fidelity, especially with the GPT-4 model.
- In a twist, I would have hoped to learn something new about LLM's intrinsic behavior by this study. Why do they mimic human performance? An abundance of strategic behavioral examples in pretraining data? Does it learn meta-reasoning that it can convert into "action" just by reading textbooks or well-known auction theories? Would a bigger, more advanced model (e.g., reasoning model) perform differently from a non-reasoning model? Would a smaller (7B/32B) model perform similarly? I am left with no such answers.

**Questions:**

Please see the weaknesses.

---

### Author Response · Authors · 2025-12-03
**Response to all reviewers**

Thank you to the reviewers for the comments. We'd like to especially thank Reviewer o6Zh for their close reading of the paper and the helpful comments.

Overall, in this submission, we made two primary contributions to the mechanism design community:
1.  Showcasing LLMs as an alternate data generating device. I.e., with a bit of prompting, LLMs can generate useful evidence on various auction designs. We think this is "mechanism-robust" in an interesting way -- a single model can generate reasonable results for many different auctions as long as we can describe the environment in code.

2. Our methods make behavioral interventions on mechanisms very cheap. We're able to test interventions like OSP, the logic of Nash deviations or prompting towards risk-aversion very cheaply. This is novel and promising -- usually in the economics literature, measuring anything towards these questions is very difficult.

We offer a general response to questions raised by the reviewers.

1) Reviewer 77vE argues the paper lacks technical novelty, citing a paper that also used LLMs to play auctions (that we also cite). This is their primary comment. We believe this sentiment completely fails to understand our contribution. Briefly, the cited paper considers a single common values environment in an English auction with multiple bidders. While we enjoyed reading this paper, there are numerous differences.
	1) In contrast to the paper, we run 1000s of auctions across a variety of settings (independent private values, affiliated private values, common values, as well as the "realistic" eBay auctions). We test LLM play across the canonical settings that auction theorists consider, replicating experiments in classic textbooks (e.g., Krishna's or Milgrom's), even replicating newer, important results like Li's OSP. The cited paper is completely different in scope and goal.
	2) Even in this setting of common value English auctions, the cited paper veers dramatically from classic settings (e.g., Milgrom-Weber 1982). Instead of drawing heterogeneous random signals, all bidders draw the same signal, mechanically fixed to $\hat{v} = 1.1 \cdot v$. This decision -- identical, constant overestimation -- mechanically (rather than strategically) induces the winner's curse and thereby obfuscates the economically interesting forces in this environment. Moreover, the decision to make the signal identical across bidders removes the strategic updating in posteriors that bidders undertake based on dropout behavior -- the key force which breaks the strategic equivalence between the English auction and the second-price sealed-bid auction in the independent private values environment. If our key interest is “can LLMs play strategic environments in a way that generates useful data as humans do,” the cited paper does not answer that question. The cited paper is explicitly not interested in auctions as mechanism, and rather just use a particular auction environment to test LLM's abilities to play sequentially.
	3) Our paper also produces a "realistic" contrast between the eBay vs. Amazon auction settings, providing synthetic data to vindicate the value of Amazon's early "soft close" rule that removed incentives to bid snipe. This insight was made explicit by academics years after the introduction of the first "soft close" rule, and belies a fundamental argument of the paper: that the ability to generate synthetic data can help move forward debates on auction design that meaningfully impact the world.
We had hoped this distinction was immediately obvious to readers of the paper.

2) There is a concern that our analysis is not robust to various prompts, models or hyperparameters. While we run simulations across a variety of prompts, it is true we don’t run simulations across a variety of models or hyperparameters. This is because our goal was not to map the space of all possible LLM play in auctions (however, it is a trivial extension to reproduce results across models and hyperparameters by changing a line in the open-source code) but rather to document how LLMs may reproduce the features that make human data useful in a variety of classic auctions, thereby demonstrating the value of synthetic data for auction design. This is the methods contribution of the paper. However, we take the point of reviewers that this may still be helpful to see, and in an updated draft of the paper, report our simulations across a variety of models and hyperparameters.
In particular, we appreciate Reviewer o6Zh’s comments on how such simulations may contribute to the paper. Reviewer dhAz asks to reproduce the setting of KL93 exactly. We already report results with prompts that match KL93 in the Appendix and show our results are robust to such prompt variation.

We thank the reviewers for their time but will be withdrawing the paper from consideration given the quality of comments. We look forward to addressing the comments of Reviewer o6Zh and resubmitting at a later time.

---

### Note · Authors · 2025-12-03

I have read and agree with the venue's withdrawal policy on behalf of myself and my co-authors.